# Global forestation and deforestation affect remote climate via adjusted atmosphere and ocean circulation

Raphael Portmann [1,3] ✉, Urs Beyerle [1], Edouard Davin[1,4], Erich M. Fischer [1], Steven De Hertog [2] & Sebastian Schemm [1]

Forests can store large amounts of carbon and provide essential ecosystem services. Massive tree planting is thus sometimes portrayed as a panacea to mitigate climate change and related impacts. Recent controversies about the potential benefits and drawbacks of forestation have centered on the carbon storage potential of forests and the local or global thermodynamic impacts. Here we discuss how global-scale forestation and deforestation change the Earth's energy balance, thereby affect the global atmospheric circulation and even have profound effects on the ocean circulation. We perform multicentury coupled climate model simulations in which preindustrial vegetation cover is either completely forested or deforested and carbon dioxide mixing ratio is kept constant. We show that global-scale forestation leads to a weakening and poleward shift of the Northern mid-latitude circulation, slows-down the Atlantic meridional overturning circulation, and affects the strength of the Hadley cell, whereas deforestation leads to reversed changes. Consequently, both land surface changes substantially affect regional precipitation, temperature, clouds, and surface wind patterns across the globe. The design process of large-scale forestation projects thus needs to take into account global circulation adjustments and their influence on remote climate.

Large-scale tree planting, along with bioenergy with carbon capture and storage, is one of the two most widely discussed land-based options for mitigating climate change complementary to fossil fuel emissions reductions[1]. At the same time, particularly in the tropics, deforestation continues at a rapid pace[2]. Both forestation [here used as an umbrella term for reforestation and afforestation] and deforestation not only affect the carbon cycle and atmospheric $CO_2$-concentrations, but also influence the local and global climate through changes in the surface energy balance (i.e., by changing albedo, evaporative fraction, and surface roughness)[3–5]. These so-called biogeophysical effects strongly depend on the location where land-use change occurs[4,6,7] and can oppose the changes in surface

temperature that result from changes in land carbon storage[6,8–11]. On a global scale, changes in albedo tend to dominate the biogeophysical temperature response[4,7]. This implies that the global mitigation effect of large-scale forestation could be diminished by the warming effect of a reduced surface albedo and altered radiative balance[8,12,13].

In addition to this direct thermodynamic warming effect, there is evidence that a massive change in forest cover also alters atmospheric dynamics, which is an important driver of regional precipitation and temperature patterns. Recent studies suggest that surface temperature changes caused by large-scale forestation or deforestation in the extratropical Northern Hemisphere can lead to a shift in the intertropical convergence zone (ITCZ) and changes in cross-equatorial

[1]Institute for Atmospheric and Climate Science, ETH Zurich, Zurich, Switzerland. [2]Department of Hydrology and Hydraulic Engineering, Vrije Universiteit Brussel, Brussels, Belgium. [3]Present address: Agroscope Reckenholz, Climate and Agriculture, Division of Agroecology and Environment, Zurich, Switzerland. [4]Present address: Wyss Academy for Nature, Climate and Environmental Physics, Oeschger Centre for Climate Change Research, University of Bern, Bern, Switzerland. ✉e-mail: raphael.portmann@alumni.ethz.ch

atmospheric heat transport[5,14–17]. Complete deforestation of the tropics may result in reduced tropical convection, which weakens the tropical source of poleward propagating Rossby waves, thereby altering extratropical circulation and surface weather[18–20]. Except for ref. 20, these studies were conducted with a slab ocean or prescribed sea-surface temperatures. However, there is evidence that ocean circulation plays an important role in shaping the climate systems' response to large-scale forestation and deforestation[4], and a change in ocean circulation may entail a reduced atmospheric circulation response[17,21]. Recent efforts based on the Land-Use Model Intercomparison Project (LUMIP) contribution to the Coupled Model Intercomparison Project Phase 6 (CMIP6) showed that several models respond to global deforestation with substantial near-surface temperature changes over the oceans[10]. In fact, the ocean thermohaline circulation appears to respond strongly to global-scale land-use change[22]. At present, it is unclear how the ocean and atmospheric circulations respond to global-scale forestation and deforestation in simulations with coupled atmosphere-ocean models. Because of their complexity, the remote effects of forestation and deforestation on weather and climate patterns remain unresolved and, inevitably, have not yet been adequately addressed in the related planning and policy-making processes. Given that a controversial recent study advocated for global-scale forestation to mitigate climate change[23] and actual plans are currently being pursued (Trillion tree campaign, www.trilliontreecampaign.org), it is important to quantify potential side-effects of large global-scale changes in forest cover on circulation and remote weather patterns.

Here we find that forestation leads to global warming of +0.5 K, which is most pronounced over northern extratropical land. Consequently, the meridional heat transport in the Northern Hemisphere decreases in the forestation simulation. The reduction manifests itself predominantly in a weakened Atlantic meridional overturning circulation (AMOC). Warming of high-latitude land surfaces results further in weaker and poleward-shifted weather systems, which, via momentum feedback to the mean flow, leads to attenuation and poleward displacement of the midlatitude westerlies. Deforestation leads to the global cooling of −1.6 K, a stronger AMOC, an accelerated extratropical jet stream, a southward shift of the intertropical convergence zone and a stronger Hadley cell in boreal winter, and a weaker Hadley cell in boreal summer. In many aspects, deforestation causes the reverse patterns compared to forestation but with larger amplitudes. These larger amplitudes are mostly related to strong snow-ice-albedo feedback in high latitudes. Because of the profound circulation changes, substantial remote changes in regional precipitation, temperature, clouds, and wind patterns occur across the globe. Key remote precipitation changes include, among others, a pronounced latitudinal shift in tropical precipitation in both simulations, and a decrease in annual mean precipitation in the Euro-Mediterranean region of more than 5% in the forestation scenario. Remote sea-surface temperature changes include a strong cooling of the North Atlantic Ocean in the forestation scenario and warming in the deforestation scenario. Large-scale forestation initiatives thus need to consider such global circulation adjustments and their effect on climate in regions remote from the forested regions.

## Results and discussion

### Impacts on temperature, near-surface winds, clouds, and precipitation

Here, simulations are performed with the Community Earth System Model (CESM) version 2, a fully coupled global climate model including a dynamic ocean model. The preindustrial control run (control) uses preindustrial land cover with ~30% of the land area covered with forests (Supplementary Fig. 1a). In the idealized forestation scenario (simulation forest), grassland, cropland, shrubs, and urban areas are turned into forests resulting in 80% forest coverage (Supplementary Fig. 1b). Conversely, in the idealized deforestation scenario (grass), the same areas are converted to grassland, i.e., forest cover is reduced to 0% and replaced by grassland (Supplementary Fig. 1c). All other aspects, including carbon dioxide ($CO_2$) mixing ratios, are kept as in control. Hence, all identified differences can be directly attributed to changes in land cover [for further details on the setup, see methods]. Our goal is to understand the fundamental effects that forestation and deforestation have on atmospheric and ocean circulation. Therefore, this idealized study is a first step toward assessing the impacts of more realistic forestation and deforestation scenarios on circulation in past, present, and future climates.

Consideration is first given to near-surface temperatures, winds, precipitation, and clouds to illustrate the profound local and remote impacts of the imposed vegetation changes on variables that are of key importance for near-surface weather and climate. Changes in regional temperature patterns may directly cause an atmospheric circulation response. Circulation changes, in turn, are of first-order importance for shaping the response of regional weather and climate to external forcing, such as land-use change or greenhouse gas emissions.

The global annual mean 2 m temperature is 0.5 °C higher in forest than in control. Most parts of northern extratropical land and the Sahel warm by more than 1 °C and some regions warm by more than 2 °C (Fig. 1a), consistent with a reduction in the clear sky surface albedo in these regions (Supplementary Fig. 2a)[4,7]. The global mean clear and full sky surface albedos decrease by 0.01. Surface warming leads to more heat days, which are defined as days with maximum temperature above 30 °C, over large parts of the midlatitude and tropical land (increase by more than 15 and locally up to 30+ days per year). Parts of the tropical land, particularly over Africa, cool and substantially less heat days occur (decrease by more than 30 days), which is a known effect of increased evapotranspiration and surface roughness[4,7]. Pronounced cooling also occurs over the North Atlantic, a phenomenon known as the North Atlantic warming hole, which is also found in greenhouse gas emission scenarios[24,25]. An approximately opposite pattern but with a larger amplitude occurs for grass (global mean cooling of −1.6 °C and global mean clear and full sky albedo increase of 0.025 and 0.026, respectively). Cooling is particularly strong over high latitudes, where it reaches well below −4 °C (Fig. 1b) and the surface albedo increases by more than 0.3 (Supplementary Fig. 2b). This strong effect is mainly linked to snow-albedo feedback over northern North America and Siberia and to a lesser extent to a sea ice-albedo feedback over the northern polar oceans (Supplementary Fig. 3). In these regions, increasing snow and sea ice cover in the summer months strongly reduces the annual mean net radiative flux at the surface. Such strong feedback is absent in forest, because there is already little summertime snow cover over northern North America and Siberia in control (Supplementary Fig. 4). For both simulations, temperature changes over northern extratropical land are larger in boreal summer (April to September) compared to winter (October to March) (Supplementary Figs. 5c, d and 6c, d). For grass, this is consistent with the albedo effect due to late spring/summertime high-latitude snow cover. For forest, however, this remains less clear and is possibly linked to reduced cloud cover in summer (Supplementary Fig. 5e).

Most albedo changes, including the effects of snow and sea ice cover, occur relatively rapidly, i.e., within the first 50 years of the simulations (Supplementary Fig. 7). Later, longer-term snow and ice-albedo feedbacks set in and contribute to further warming and albedo reduction in the forestation scenario and cooling and albedo increases in the deforestation scenario. To compare the magnitudes of the imposed radiative forcings with the forcing from historical anthropogenic $CO_2$-emissions, a simplified expression is used to translate a global mean top-of-atmosphere radiative forcing to changes in atmospheric $CO_2$-concentrations (see methods). The initial forcings (first 5 years) are 0.70 W m$^{-2}$ (corresponding to about 40 ppm) in forest and −0.85 W m$^{-2}$ (−42 ppm) in grass. The magnitude of the forcings in both

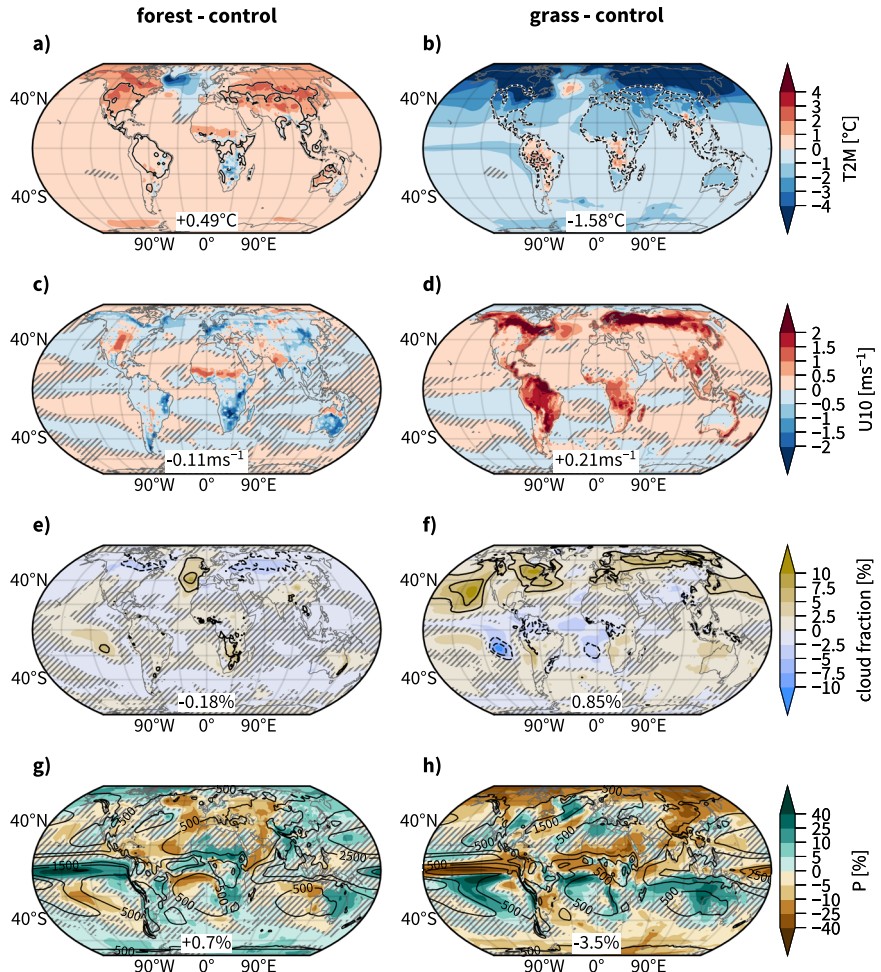

**Fig. 1 | Changes of variables relevant to near-surface weather and climate.** Panels (**a**, **c**, **e**, **g**) show changes in *forest* and **b**, **d**, **f**, **h** changes in *grass* compared to *control*. Variables shown are (**a**, **b**) absolute changes of annual mean 2 m temperature (T2M, shading, in °C) and the number of days with a maximum temperature above 30 °C (heat days, +15 days in solid contours and −15 days dashed) and **c**, **d** absolute changes of annual mean 10 m wind speed (shading, in ms⁻¹),

**e**, **f** absolute changes of annual mean total cloud fractions (shading, in %) and low cloud fractions (black contours, shown are −10, −5, 5, and 10% contour levels), and **g**, **h** relative changes in annual mean precipitation (shading, in %, note the nonlinear color spacing). Statistically insignificant results are hatched, see Methods section. This Figure was produced using Cartopy, which is licensed under the GNULesserGeneralPublicLicense.

simulations is, therefore approximately equivalent to one-third of the increase in atmospheric $CO_2$-concentration since 1850 (126 ppm). During the first years, the forcing in *forest* decreases (average of 0.57 W m⁻² in years 5–10) and becomes more negative in *grass* (average of −0.91 W m⁻² in years 5–10). The reason for the initial negative forcing trend in *grass* is the rapid snow and ice-albedo feedback. At the end of the simulations (years 296–300), the forcings have reduced to 0.41 W m⁻² (−22 ppm) in *forest* and −0.35 W m⁻² (−18 ppm) in *grass*, which shows that radiative equilibrium is not yet reached.

Near-surface wind speeds decrease over most land areas in *forest* and increase in *grass*, which agrees with the changes in surface roughness (Fig. 1c, d). However, in *forest* notable exceptions with increasing near-surface winds occur over central North America, the Sahel, northern India, and northern Australia, which are all strongly forested regions. This is likely related to regional-scale circulation changes related to orography, land-sea contrasts, or thermally-driven circulation contrasts between the desert and vegetated areas. Further, near-surface winds in the North Atlantic warming-hole region weaken *forest* and strengthen *grass*.

Global mean cloud cover decreases in *forest* and increases in *grass* (Fig. 1e, f). Over northern extratropical land cloud cover is reduced by more than 2.5% in *forest* and increased by up to 10% *grass* and changes are more pronounced in boreal summer compared to winter (for

seasonal changes, see Supplementary Figs. 5e, f and 6e, f). This partly contradicts recent observational evidence[26,27] but is in agreement with a modeling study[16] which argues that, after a certain degree of midlatitude forestation, increased sensible heat fluxes lead to warming and drying of the troposphere and thereby inhibit cloud formation. Over tropical land, changes in cloud cover are less pronounced and their sign depends on the region. Cloud cover changes also occur over oceans, pointing towards changes in ocean circulation and ocean–atmosphere interactions but also changes related to the mid-latitude storm tracks. In *forest*, cloud cover increases particularly strongly over the eastern North Atlantic [up to 7.5%, mostly in boreal summer] and the tropical South Pacific [up to 5%] (Fig. 1e). In *grass*, cloud cover increases particularly strongly over the eastern North Pacific and the western North Atlantic [5–10%] and decreases over a relatively confined region over the eastern tropical South Pacific [up to 10%] (Fig. 1f). In both experiments, low-clouds contribute strongly to the response.

There are also profound consequences on annual mean precipitation (Fig. 1g, h), with global mean precipitation increasing in *forest* by 0.8% and decreasing in *grass* by 3.0%. The spatial patterns of the sign of precipitation changes in both *forest* and *grass* are qualitatively consistent with end-of-century precipitation changes in CMIP5 global warming projections (with a reversed sign for *grass*)[28]. This

suggests that, to first order, mean precipitation changes are the result of global warming/cooling induced by land cover change. The percentage change per degree of temperature change is 1.6% K$^{-1}$ in forest and 1.90% K$^{-1}$ in grass, which is smaller than in CO2-driven global warming scenarios in CMIP6 models (2.1–3.1% K$^{-1}$)[29]. Precipitation decreases in the Euro-Mediterranean region in forest and increases in grass by 10% and more. In both simulations, relative changes are larger in boreal summer than in winter (Supplementary Figs. 5g, h and 6g, h). Tropical precipitation changes are characterized by latitudinal shifts which depend remarkably little on the seasons. Exceptions with pronounced seasonal differences are the equatorial North Atlantic/West Africa (much larger relative increase in boreal winter) in forest and the western tropical Pacific (stronger decrease in boreal summer) and East Africa (changes occur mostly in boreal summer) in grass.

Regions with strong precipitation and cloud cover changes do not necessarily coincide with large changes in forest cover. This indicates that the large-scale circulation adjustments are underlying the changes in precipitation and cloud cover. These large-scale circulation adjustments result from the changes in the Earth's energy balance and are the focus of the subsequent sections.

## Changes in the meridional heat transport in the climate system

A necessary consequence of the surface radiative forcing that drives pronounced warming and cooling over northern extratropical land are changes in the meridional heat transport in the climate system (Fig. 2), which mainly occur in the ocean (Fig. 2b, c) in both simulations. In forest, the total poleward heat transport is reduced in the Northern Hemisphere (peak transport decreases by ~5%) and, to a lesser extent, increased in the Southern Hemisphere (+2%) (Fig. 2b). The reduction in the Northern Hemisphere is solely due to a reduction in poleward ocean heat transport by 10–25% between ~10 and 60°N relative to control. In the atmosphere, the decreased dry heat transport is almost compensated by an increased latent heat transport, which is a result of the general moistening of the atmosphere everywhere. In grass, the total poleward heat transport increases by approximately 10% (peak transport) in the Northern Hemisphere and decreases by ~3% in the Southern Hemisphere. The changes are also dominated by ocean heat transport (an increase of 20–50% between 10 and 60°N). However, atmospheric heat transport also increases by ~5% (peak transport) in the Northern Hemisphere, mainly due to increased dry heat flux.

A major contributor to meridional ocean heat transport is the Atlantic meridional overturning circulation (AMOC), which transports warm surface water from the tropics along the Gulf stream into the extratropical North Atlantic, where it cools and eventually sinks and recirculates southward at the bottom of the ocean basin[30]. A change in the strength of the AMOC directly explains a change in ocean heat transport. In forest, the AMOC weakens and becomes shallower and in grass it strengthens and becomes deeper across all latitudes between

30°S and 60°N (Fig. 3). To diagnose changes in water mass transformation, AMOC is also computed in potential density coordinates (Supplementary Fig. 8). In this framework, the AMOC maximum and also the strongest simulated AMOC changes occur further north (between 40–60°N) than in height coordinates. This points to the key role of changes in water mass transformations in the subpolar North Atlantic for the changes in AMOC strength. When quantified with the maximum AMOC index between 20 and 70°N (see methods), the mean annual AMOC strength decreases by 22% in forest and increases by 49% in grass. A weakening of the AMOC and the emergence of a North Atlantic warming hole as occurs in forest emerges also in global warming scenarios driven by greenhouse gas emissions[31].

Why does ocean heat transport respond so strongly in the forestation scenario while atmospheric heat transport changes only marginally, even if the initial perturbations occur over land? In fact, at the beginning of the simulations (first 40 years), the atmosphere and ocean contribute equally to the reduced heat transport in the midlatitudes (see Supplementary Fig. 9). During the following decades, the strength of the AMOC decreases strongly and the ocean takes over the bulk of the change in meridional heat transport. Initially, forestation increases surface sensible heat flux over northern extratropical land due to lower albedo and higher surface roughness (Supplementary Fig. 11a) and increases near-surface air temperature. When this warmer air is advected over the adjacent ocean, sensible heat fluxes over the ocean decrease due to a reduced air-sea temperature contrast (see Supplementary Fig. 11b). This is further supported by the weaker near-surface winds, which also contribute to reduced ocean–atmosphere latent heat fluxes. At the same time, longwave cooling of the atmosphere over extratropical oceans is enhanced due to warmer air temperatures. As a consequence, the net energy fluxes into the atmosphere over the extratropical oceans decrease. In total, the extratropical atmosphere between 40 and 70°N receives more additional energy over land than it loses over the oceans during the first four decades of the simulation (see Supplementary Fig. 11c). Given the net negative changes in energy input to the atmosphere further equatorward between 15 and 40°N, the meridional atmospheric heat transport must decrease.

The atmospheric heat budget over land remains fairly unchanged over time, whereas over the northern extratropical oceans between 45 and 70°N, latent heat fluxes strongly reduce during the first 120 years (Supplementary Fig. 11b, e, h), eventually fully compensating for the enhanced sensible heat fluxes over northern extratropical land. As a result, atmospheric heat transport increases again to its original magnitude. This drastic reduction in ocean surface heat fluxes is caused by ocean cooling in the northern extratropics (i.e., the emergence of the North Atlantic warming hole), likely due to the slowdown of the AMOC[25,31]. The strong response in ocean heat transport during the first century of the simulation reduces the initial response in

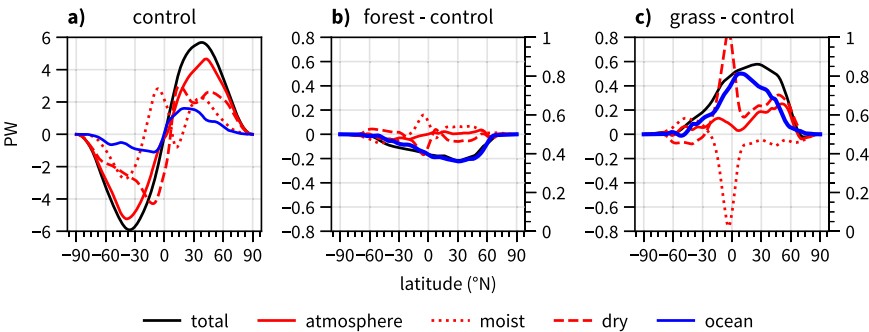

**Fig. 2 | Changes of the annual mean meridional heat transport in atmosphere and ocean.** Shown are **a** meridional heat transport in control, **b** its changes in forest with respect to control, and **c** its changes in grass with respect to control. The total

heat transport (black) is separated into atmospheric (red) and ocean (blue) transport. Atmospheric transport is further separated into the transport of moist (dotted) and dry static energy (dashed).

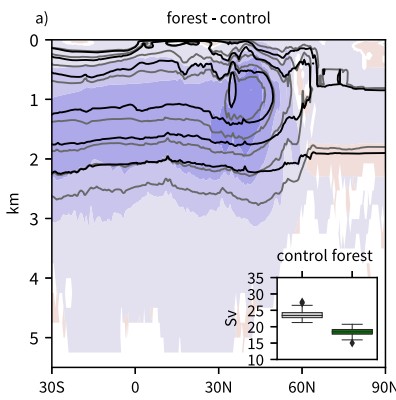
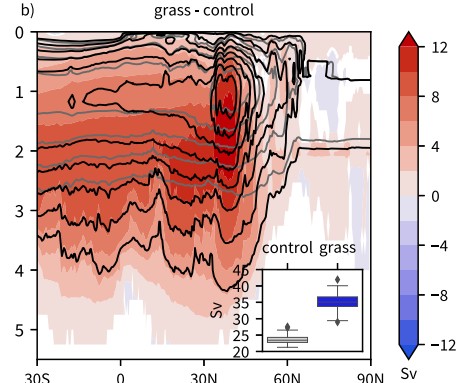

**Fig. 3 | Slowdown and acceleration of the Atlantic meridional overturning circulation.** Shown is the annual mean meridional mass stream function in the Atlantic, i.e., the AMOC (contours, in Sv, only values above 4 Sv shown with a 4 Sv interval) for **a** forest (black) and control (gray) and **b** grass (black) and control (gray), as well as differences of experiments relative to control (shading, in Sv). Statistically insignificant differences are shown in white, see Methods section. Insets in the bottom right of each panel show standard box plots of the annual mean AMOC index (in Sv) for **a** forest and control and **b** grass and control.

atmospheric heat transport to quasi-zero after roughly a century (Supplementary Fig. 9e).

The critical role that the oceans, and particularly the Atlantic Ocean, play in the climate system's response to forestation and deforestation is further supported by the substantial increase in ocean heat content in *forest* and its decrease in *grass* in the mid- and upper oceans (with the exception of the North Atlantic warming-hole region, Supplementary Fig. 12). In the forestation simulation, the ocean heat content decreases in the deep ocean, likely due to the shallower and weaker AMOC, which transports heat less efficiently to the deep ocean.

### Changes in midlatitude westerlies, weather systems and the extratropical jet stream

The warming in the forestation scenario and the cooling in the deforestation scenario in the extratropics extend through the entire depth of the troposphere and temperature changes are most pronounced in middle latitudes in *forest* and in middle and polar latitudes in *grass*. The pole-to-equator temperature gradient adapts according to the imposed warming and cooling (Supplementary Fig. 13b, c). The Northern Hemisphere time means westerly flow weakens between 30–50°N in *forest* and strengthens between 20–60°N in *grass*, in agreement with what can be expected from the thermal wind relation[32]. The changes are particularly pronounced around the upper-tropospheric jet stream. The midlatitude westerlies are longitudinally asymmetric due to planetary-scale stationary Rossby waves excited by orography, thermal land-sea contrasts and the climatological distribution of diabatic forcing[33,34]. This stationary wave pattern is shaped by troughs over eastern North America and eastern Asia (Fig. 4a).

Continental-scale forestation and deforestation alter stationary wave patterns. For example, over North America, surface warming in the forestation scenario weakens the mean trough, and the adjacent climatological pressure gradient decreases, while surface cooling in the deforestation scenario produces the opposite result (Fig. 4c, e). The dynamics of these stationary waves are well described by barotropic theory[33,35] and the response of upper-level westerlies in the vicinity of continental-scale troughs and ridges is well aligned with the change in these stationary Rossby wave patterns and large-scale pressure gradients. Additional modification of stationary waves may arise from changes in tropical Rossby wave sources. A closer inspection of changes in tropical Rossby wave sources[36], however, reveals no well-marked changes between the two scenarios and the control experiment (Supplementary Fig. 14).

Over the main storm track regions, the intensity and position of the jet stream are driven by transient baroclinic eddies (i.e., midlatitude weather systems). These so-called eddy-driven jets are the

result of eddy momentum flux convergence[37–39], which is inherently connected to the life cycle and propagation of synoptic-scale weather systems[40]. Eddy-driven deep jets are characterized by enhanced baroclinicity throughout the troposphere, hence the term deep jets. They are common over the oceanic storm track region, while the thermally-driven shallow jets are common over the subtropics[41]. In *forest*, the deep jets migrate poleward in both hemispheres, whereas in *grass* the jets retreat equatorward in the Southern Hemisphere and do not shift in the Northern Hemisphere (Fig. 4d, f). The dynamics of eddy-driven deep jets are well described by the orientation and divergence of the **E**-vector[40], which indicates zonal momentum forcing by synoptic-scale weather systems and the preferred type of Rossby wave breaking that displaces the eddy-driven jet[42,43]. **E**-vector divergence indicates flow acceleration, for example, over the western Gulf Stream region (Fig. 4b). In the forestation scenario, reduced **E**-vector divergence in this region suggests weaker eddy momentum flux convergence and thus reduced westerlies which agrees with a reduced feature-based deep jet detection frequency and is in line with weaker eddies over the western North Atlantic (Supplementary Fig. 15a). The momentum flux divergence anomalies over wider parts of the Southern Hemisphere are in agreement with the overall latitudinal displacement of the deep jets in both scenarios, indicating a shift in midlatitude storm tracks (poleward in *forest* and equatorward in *grass*). In the forestation scenario, a poleward shift is also discernible in the Northern Hemisphere, where the relative changes are locally even larger. This agrees with results from comprehensive $CO_2$-driven global warming simulations[44,45]. A poleward shift of the storm tracks has been previously linked to increased upper-tropospheric baroclinicity as a potential result of global warming[46,47], to the widening of the Hadley circulation[48] and to a stronger poleward propagation of weather systems[49]. There is a pronounced seasonality in the response of the extratropical jet stream in *forest*, particularly in the Northern Hemisphere (Supplementary Figs. 16c, 17c). Absolute changes are larger in boreal winter than in boreal summer. Further, the North Pacific jet stream predominantly shifts poleward in boreal winter and weakens in boreal summer. In the deforestation scenario, the changes in the mean midlatitude circulation patterns are generally reversed and stronger than those found in the forestation scenario, including enhanced eddy kinetic energy (Supplementary Fig. 15b), stronger westerlies (Fig. 4e), stronger eddy momentum flux convergence and more frequent deep jets (Fig. 4f), but without latitudinal shift of the jet stream in the Northern Hemisphere. The jet intensifies primarily over the western North Atlantic in boreal winter and across the whole northern midlatitudes in boreal summer (Supplementary Figs. 16e, 17e).

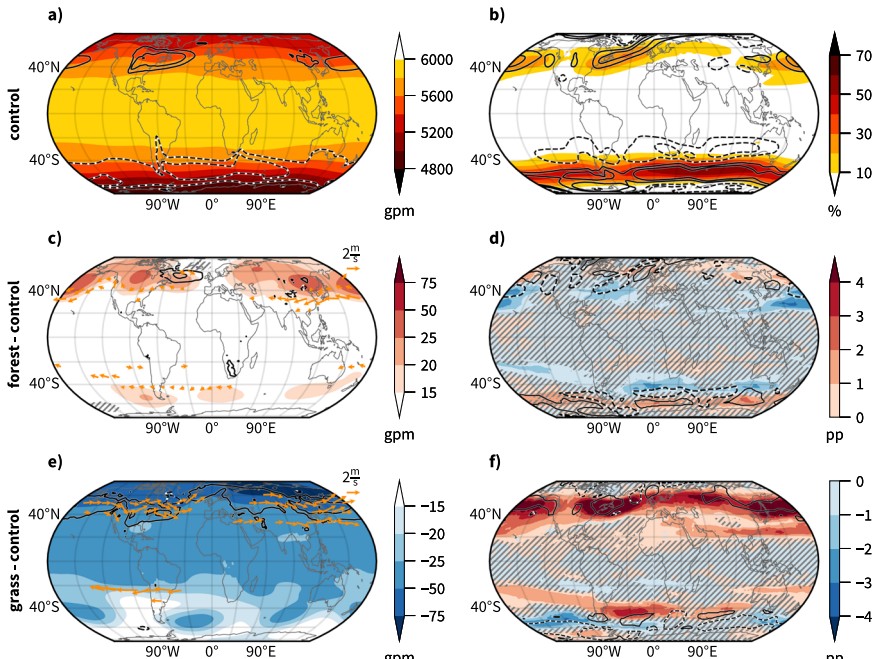

**Fig. 4 | Changes in the extratropical atmospheric circulation. a** Annual mean geopotential height at 500 hPa (Z500, shading, in gpm) and meridional component of the eddy heat flux at 850 hPa ($\overline{v'T'}$, contours, −15, −10, 10, and 15 K ms$^{-1}$) in simulation *control*, **b** annual mean frequencies of deep jets (shading, in %) and E-vector divergence (contours, −3, −1.5, 1.5 and 3 m$^2$ s$^{-2}$(100 km)$^{-1}$), **c, e** differences in annual mean Z500 (shading, in gpm), eddy heat flux at 850 hPa (contours, −3, −1, 1, and 3 K ms$^{-1}$), and representative wind differences at 300 hPa (arrows, only the sixty globally largest differences at every tenth latitude and longitude index are shown, i.e., if larger than 0.6 ms$^{-1}$ for *forest* and 0.6 ms$^{-1}$ 1.4 ms$^{-1}$ int *grass*) for **c** forest minus *control* and **e** *grass* minus *control*, and **d**, **f** differences of annual mean

deep jet frequencies (shading, in percent points [pp]) and E-vector divergence (contours, −0.15 and 0.15 m$^2$ s$^{-2}$(100 km)$^{-1}$, only shown for regions where E-vector divergence is larger than 0.5 m$^2$ s$^{-2}$(100 km)$^{-1}$ in *control*) for **d** *forest* minus *control* and **f** *grass* minus *control*. For all contours, positive values are solid and negative values dashed. Statistically insignificant changes (see methods) are hatched for variables shown in shading. For other variables, only statistically significant results are shown, except for the changes in E-vector divergence. These fields would otherwise be too patchy in *grass* and are not significant in *forest* according to the test used in this study. This Figure was produced using Cartopy, which is licensed under the GNULesserGeneralPublicLicense.

---

A noticeable anomaly in both scenarios, which is more pronounced in *forest*, is the behavior of the eddies over the central to eastern North Atlantic and along the southern tip of Greenland (Fig. 4c–f). Despite the hemispheric-wide reduction in the mean baroclinicity, which reduces the potential for baroclinic eddy growth[50,51], the eddy heat flux is increased in the forestation scenario (and vice versa in the deforestation scenario) over the North Atlantic warming-hole region (Fig. 4c, e). The regional decrease in sea-surface temperatures related to the slowdown in the Atlantic meridional overturning circulation (Fig. 3) drives a local atmospheric temperature and circulation anomaly, which enables enhanced baroclinic growth of synoptic-scale eddies and subsequently eddy kinetic energy (Supplementary Fig. 15a) and eddy momentum flux convergence downstream that ultimately shifts the eddy-driven jet over the northeastern North Atlantic poleward (Fig. 4d)[52] despite the hemispheric-wide decrease in near-surface baroclinicity. This effect is more pronounced in boreal winter but occurs also in boreal summer (Supplementary Figs. 16c, 17c). In the deforestation scenario, this anomalous eddy forcing over the North Atlantic is too weak relative to the overall enhanced westerly winds to be detectable as a jet shift.

Overall, the northern midlatitude atmospheric circulation responds with a weakening and poleward displacement to global-scale forestation and with strengthening to deforestation. The southern midlatitude circulation mainly reacts with a poleward shift to forestation and an equatorward shift to deforestation. The latitudinal shifts in the storm tracks agree well with the latitudinal shift in the annual mean precipitation, particularly over the North Pacific and in the Southern Hemisphere (Fig. 1e, f). We must expect related albeit weaker changes in midlatitude weather under regionally confined large-scale mid- and high-latitude forestation and deforestation scenarios.

## Changes to the ITCZ and the Hadley cell

The Hadley cell is one of the key atmospheric circulation features that redistributes heat and momentum in the climate system. Changes in Hadley cell position, strength, or width directly impact tropical and subtropical precipitation patterns. Given its strong seasonality, the Hadley cell is best diagnosed separately from October to March (boreal winter) and April to September (boreal summer).

In the simulation, the *forest* Northern Hemispheric cell weakens in boreal winter (Fig. 5a, relative changes of about 2–4%). This is consistent with the weaker equator-to-pole and hemispheric temperature contrasts due to the warming in the Northern extratropics. There are also signs of an intensification of the Southern Hemispheric cell in boreal summer by the same order of magnitude (Fig. 5c). These modest circulation changes are in agreement with the weak changes in total atmospheric heat transport (Fig. 2b).

In the deforestation scenario, the circulation changes are more pronounced. In boreal winter, the Northern Hemispheric cell strengthens and broadens on its southern flank (10–90% increase), while the Southern Hemispheric cell narrows on its northern flank. This results in a southward shift of the zero isoline of the meridional mass stream function, which is a proxy for the location of the ITCZ. The southward shift of the ITCZ is also evident from the annual mean meridional energy flux perspective, which identifies the location of the ITCZ as the latitude where the meridional atmospheric energy flux changes sign (energy-flux-equator, see ref. 53). The southward shift of the energy-flux-equator is a direct result of the increased meridional energy transport at 0°N (Supplementary Fig. 18), which in turn arises due to the increased atmospheric energy transport out of the tropics at approximately 30°N (Fig. 2c). This is consistent with the increased eddy heat fluxes in Fig. 4e). In conclusion, cooling in the northern

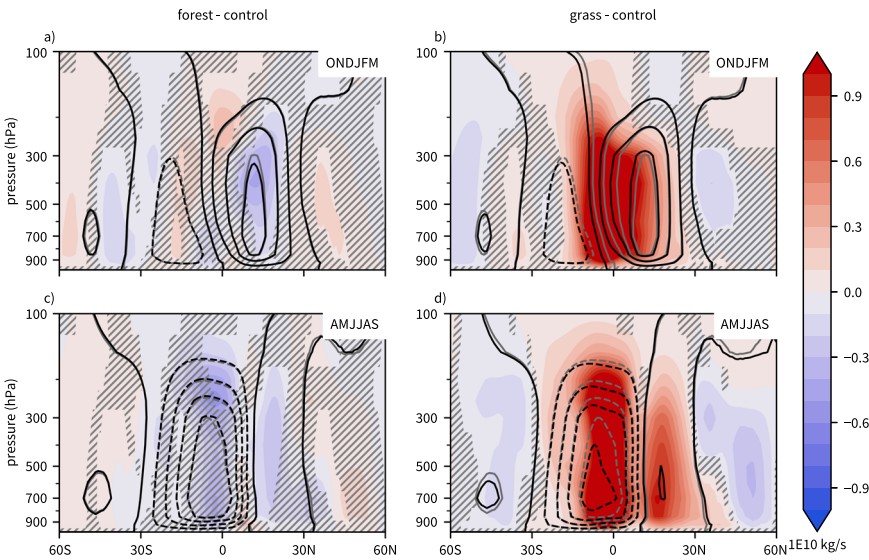

**Fig. 5 | Seasonal changes in the strength of the Hadley cell.** Seasonal mean meridional mass stream function (contours, every 2E10 kg s⁻¹) for (**a, c**) forest (black) and control (gray) and **b, d** grass (black) and control (gray), as well as differences of both experiments relative to the control (shading, in 1E10 kg s⁻¹) for (**a, b**) October to March and **c, d** April to September. Statistically insignificant differences are hatched, see Methods.

extratropics leads to increased meridional (eddy) heat flux out of the subtropics which is compensated by increased heat transport by the Hadley cell. This is achieved by intensification and southward extension of the Northern Hemispheric cell further into the warmer Southern Hemisphere.

In boreal summer, the Southern Hemispheric branch of the Hadley cell weakens in the deforestation simulation (Fig. 5d) as a result of the reduced inter-hemispheric temperature contrast (because of the cooler summer in the Northern Hemisphere). Less energy has to be transported from the Northern to the Southern Hemisphere. Because the ITCZ is located in the Northern Hemisphere in boreal summer, this is consistent with a weakening of the Southern Hemispheric branch of the Hadley cell.

The poleward momentum transport of the Hadley cell results in thermally-driven shallow jet streams in the subtropics[39]. The intensity changes of the Hadley cell, therefore directly affect the strength of these jet streams (Supplementary Fig. 19). As a result, the thermally-driven jet over Asia weakens in *forest* and strengthens in *grass* and the thermally-driven jet over South America intensifies in *forest* and weakens in *grass*.

The expected annual mean net effect on tropical precipitation from the dynamic changes is consistent with the simulated changes in tropical annual mean precipitation, which shows a northward shift in forest and a southward shift in *grass* (Fig. 1g, h). In *forest*, the weakening of the Northern Hemispheric branch results in less precipitation just south of the equator (i.e., at the ITCZ) and the strengthening of the Southern Hemispheric branch leads to a precipitation increase north of the equator. Similarly, the weakening of the Southern Hemispheric branch of the Hadley cell, the strengthening of the Northern Hemispheric branch and the southward shift of the ITCZ in *grass* contribute to precipitation increases south of the equator and decreases north of the equator.

**Implications for future forestation strategies**

Current forestation strategies overly neglect or underestimate potential impacts on atmospheric and ocean dynamics because of the focus on thermodynamic effects from carbon storage. From this study, it appears that changes in the Earth's energy budget due to global-scale forestation and deforestation may entail large-scale circulation changes in the atmosphere and ocean. This results in changes in remote climate patterns beyond the known local and global thermodynamic

effects, which in turn are known to partially offset the cooling or warming effects of carbon sequestration[12,13]. The here identified circulation changes are profound and affect the midlatitude westerlies and the Hadley circulation among others; therefore, even with regionally limited or less dense forestation, remote circulation changes must be expected. Hence the design process of future large-scale forestation projects needs a comprehensive assessment of biogeophysical consequences, including remote effects via changes in atmospheric and ocean circulation.

In particular, the strong response of the AMOC to forestation and deforestation is of utmost importance, as the AMOC is predicted to slowdown under anthropogenic global warming[54]. This ocean response has previously been largely overlooked due to missing atmosphere-ocean coupling. The higher surface temperatures and reduced 10-m wind speeds following forestation reduce ocean surface heat fluxes. It is plausible that this leads to initial warming and freshening from reduced heat and water loss in the North Atlantic. As a result, one would expect the density of the near-surface water to decrease, which inhibits deep water formation in the Labrador Sea. Regional atmospheric circulation changes related to surface cyclones and cold air outbreaks[55,56], as well as changes to the wind-driven ocean gyre circulation[25], may be crucial elements in such a process chain.

Our results are likely sensitive to model type and land-surface parametrisations. A recent comparison of the response of nine CMIP6 models to global deforestation indeed shows remarkable variability in the 2 m temperature response. Hence, similar variability in the atmospheric circulation response can be expected. However, next to CESM2, two of these models produce local warming over the North Atlantic, indicating a similar response of the ocean circulation in these models[10].

As a next step, research is needed to assess the mechanisms described here under more complex model configurations, e.g., considering different scales of forestation[16], focusing on individual regions[18], including transient future climate scenarios[57], and adding biogeochemical effects[10].

Finally, it is important to emphasize that our study must not be used as a general argument against forestation given the obvious positive merits (air quality, biodiversity, nutrition, local cooling through evapotranspiration and shading, protecting water resources, recreation among others), with tropical forestation being established as clearly beneficial. But the remote circulation effects on regional

climates described here should be taken into account and carefully evaluated to maximize the benefit of forestation efforts at any latitude and scale.

## Methods

### Simulations

Our simulations are performed with the Community Earth System Model (CESM) version 2.1.2[58]. The model was running in fully coupled mode, including the Community Atmosphere Model (CAM6, 32 vertical levels) and the Community Land Model (CLM5), the Parallel Ocean Program version 2 (POP2, 60 vertical levels), the Los Alamos National Laboratory Sea Ice model (CICE5), and the hydrological routing model Model for Scale Adaptive River Transport (MOSART) all at about 1° horizontal resolution. The climate forcing was set to preindustrial conditions and remained constant throughout the full simulation period. In the standard preindustrial experiment, the first 200 years were excluded to ensure that the simulation is equilibrated. After these 200 years, the forestation and deforestation experiments, as well as the control simulation, were branched off and run for 300 years.

The land cover maps for the forestation and deforestation simulations were created following the approach developed in ref. 59. In the forestation experiment, all plant functional types (including urban land) except forest and bare soil are removed from the preindustrial land cover maps. Then the bare soil fraction is kept constant while the forest plant functional types at each grid point are upscaled to fill up the entire remaining land area fraction at each grid point. This approach ensures that the preindustrial distribution of tree functional types is retained. At grid points where the preindustrial forest area fraction is zero, the zonal mean distribution of tree functional types is used. As a consequence of this approach, regions with large fractions of bare soil (e.g., deserts) do not exhibit large changes. Accordingly, in the deforestation experiment, the same approach is taken but with retaining the grass instead of the forest plant functional types.

To remove the effects of the initial adjustments of the model to the land-surface perturbation, the data were evaluated for years 50 to 300 after, unless stated otherwise. The five-decade-long spin-up period is justified by the observation that this is approximately the period during which the initial rapid adjustment of parameters such as the 2 m temperature, albedo and snow/sea ice fractions takes place (Supplementary Fig. 7). It is important to note that both experiments are not yet in full equilibrium after these five decades and even not after three centuries, which can be inferred from the radiative forcing estimations. Excluding the first 100 years from the analysis results in qualitatively similar results. A spin-up period of five decades for the initial adjustment period is in line with the LUMIP/CMIP6 experiments on global deforestation, which omit the first three decades after linear deforestation over 50 years[10,60].

### Statistical significance testing

We apply a two-sided Wilcoxon rank-sum test to ensure that the differences of annual or seasonal means between each of the two experiments and the control simulation are robust[61]. Here, the null hypothesis is tested that, at a certain grid point, it is equally likely that the value of the considered variable in a randomly selected year or season in the experiment is larger or smaller than in a randomly picked year or season in the control simulation. The test data for each experiment consist of annual or seasonal means values of each year from 50 to 300. As this test is applied to fields, $p$ values are corrected using a Benjamini–Hochberg correction[62]. The false discovery rate is set to a value of $\alpha_{fdr} = 0.05$ and differences are only shown for regions where the null hypothesis is rejected at this level of $\alpha_{fdr}$.

### Radiative forcing estimation

Here, we describe a simple approach to compare the net radiative forcing found in our simulations to the radiative forcing by a given change in atmospheric $CO_2$-concentrations. Using a simplified expression which has shown to agree well with explicit radiative transfer calculations, the radiative forcing $\Delta F$ of an increase in atmospheric $CO_2$ from a mixing ratio $C_0$ to $C$ can be calculated as[1]

$$\Delta F = \alpha \ln\left(\frac{C}{C_0}\right) \quad (1)$$

with $\alpha = 5.35$. This expression can be used to compute the change of atmospheric $CO_2$ since preindustrial values that would be required to achieve the same radiative forcing as in our simulations, i.e. an equivalent $CO_2$ forcing, as

$$\Delta C = C_0 \exp\frac{\Delta F}{\alpha} - C_0 \quad (2)$$

This allows for a comparison between our imposed global mean radiative forcing and the forcing due to anthropogenic global warming. Anthropogenic emissions have caused an increase in the annual mean atmospheric $CO_2$ mixing ratio from 284 ppm in 1850[63] to 410 ppm in 2020[54], which corresponds to a $\Delta C$ of 126 ppm. We compute the top-of-atmosphere net radiative imbalance at the beginning (average of the first 5 years) and end (average of the last 5 years) of the simulations *forest* and *grass*. The difference to the climatological mean top-of-atmosphere net radiation in *control* is then considered as the radiative forcing. The radiative forcing at the beginning can then be interpreted as the initial forcing, including very rapid adjustments. The radiative forcing at the end includes long-term adjustments and feedbacks.

### Identification of jet streams

Jet streams are identified as instantaneous regions with enhanced wind speeds in the tropopause region[41] based on the 6-h model output. This method is based on the vertical average of the total wind speed between 400 and 100 hPa defined as

$$\alpha\text{vel} = \frac{1}{400\,\text{hPa} - 100\,\text{hPa}} \int_{100\,\text{hPa}}^{400\,\text{hPa}} (u^2 + v^2)^{\frac{1}{2}} dp \quad (3)$$

where $u$ and $v$ are the zonal and meridional wind components, respectively. Jet streams are then identified as regions with $\alpha\text{vel} \geq 30$ ms$^{-1}$. Further, the jet streams are separated into deep and shallow jets based on an index of upper-tropospheric wind shear

$$\Delta v_{rel} = \frac{\mathbf{v}_{200} - \mathbf{v}_{500}}{\mathbf{v}_{200}} \quad (4)$$

where $\mathbf{v}_{200}$ and $\mathbf{v}_{500}$ denote the horizontal wind speeds at 200 and 500 hPa, respectively. Deep jets exhibit low-upper-tropospheric wind shear ($\Delta v_{rel} < 0.4$) and hence are accompanied predominantly by lower-tropospheric baroclinicity. They can therefore be considered as manifestations of the extratropical or eddy-driven jet stream. For shallow jets, baroclinicity is confined to the upper troposphere ($\Delta v_{rel} > 0.4$) and they are considered manifestations of the subtropical or thermally-driven jet stream. For the more detailed rationale behind this method and the selection of the thresholds for $\alpha\text{vel}$ and $\Delta v_{rel}$, the reader is referred to[41].

### E-vector

The effect of transient extratropical eddies on the mean flow can be assessed with the **E**-vector, which was introduced by ref. 40 and

modified by ref. 64 to the form used in this study

$$\mathbf{E} = \left[\frac{1}{2}\overline{(v'^2 - u'^2)}, -\overline{u'v'}\right]. \qquad (5)$$

The overbar denotes the time average and u' and v' are the zonal and meridional transient flow anomalies. They are obtained by subtracting the 10-day high-pass filtered flow component from the 6-h wind fields. Regions of **E**-vector divergence denote the acceleration of the mean flow due to transient eddy activity and **E**-vector convergence denotes deceleration.

### Meridional heat transport

The total meridional heat transport (MHT) at latitude $\phi$ is calculated based on the climatological energy balance at the top of the atmosphere $R_{TOA}$[65].

$$\text{MHT}(\phi) = -2\pi a^2 \int_{\phi}^{90} \cos(\phi)\left[R_{TOA}(\phi) - R_{TOA}^{imbal}\right]d\phi. \qquad (6)$$

$R_{TOA}^{imbal}$ is the global mean value of $R_{TOA}$, which, in practice, is non-zero and has to be subtracted from $R_{TOA}(\phi)$ in order to ensure energy conservation (for details, see ref. 65). In a similar fashion, atmospheric heat transport (AHT) is computed as

$$\text{AHT}(\phi) = -2\pi a^2 \int_{\phi}^{90} \cos(\phi)\left[E_{atm}(\phi) - E_{atm}^{imbal}\right]d\phi, \qquad (7)$$

where $E_{atm}$ is the net energy input into the atmospheric column computed as the sum of net TOA radiation and net surface energy fluxes ($E_{sfc}$), which is the sum of net radiative fluxes at the surface ($R_{sfc}$), sensible and latent heat fluxes (SH and LH), and latent heat fluxes due to snow melt at the surface (SNOW) [all fluxes are positive if net energy input is into the atmosphere], i.e., $E_{atm} = R_{TOA} + E_{sfc} = R_{TOA} + R_{sfc} + \text{SH} + \text{LH} + \text{SNOW}$. The transport of latent heat (i.e., the moist component of atmospheric energy transport) is computed as

$$\text{AHT}_{moist}(\phi) = -2\pi a^2 \int_{\phi}^{90} \cos(\phi)[\text{LHnet}_{atm}(\phi) - \text{LHnet}_{atm}^{imbal}]d\phi, \qquad (8)$$

with $\text{LHnet}_{atm} = \text{LH} - P\rho_w L_{vap}$ denoting the net input of latent heat into the atmospheric column ($\rho_w$: density of water, $L_{vap}$: specific heat of vaporization). The dry component of the atmospheric transport and the ocean heat transport are finally computed as residuals.

### AMOC index

The AMOC index is a measure of how much warm surface water is transported poleward in the North Atlantic at a given latitude and is defined as the maximum of the meridional overturning stream function below 500 m. Here, we use the maximum AMOC index between 20–70°N to avoid the sensitivity of the results to the selection of a particular latitude. This approach is similar to ref. 31.

### Data availability

The data to reproduce all Figures in this manuscript and the supplement, the annual mean data behind all five Figures in the manuscript, as well as the modified land cover data were available for download in ETH Zurich's Research-Collection repository https://www.research-collection.ethz.ch/handle/20.500.11850/556688. ETH Zurich's Research-Collection adheres to the FAIR principles. The full simulation data were stored at the Institute of Atmospheric and Climate Science, ETH Zurich, for at least 10 years and are available on request.

### Code availability

The code to produce the modified land-surface data were available at the same ETH Zurich research-collection repository https://www.research-collection.ethz.ch/handle/20.500.11850/556688.

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

## Acknowledgements

The authors thank Heini Wernli and Reto Knutti (ETH Zurich) for helpful guidance on the project design and comments during the writing process. The computations were performed on the Euler cluster at ETH Zurich, supported by the H2020 European Research

Council (project INTEXseas; grant no. 787652). S.S. is supported by funding from the European Research Council (ERC) under the European Union's Horizon 2020 research and innovation program (grant agreement no. 848698). S.D.H. is supported by the LAMACLIMA project, part of AXIS, an ERA-NET initiated by JPI Climate, and funded by BELSPO (BE, Grant No. B2/181/P1) with co-funding by the European Union (Grant No. 776608).

## Author contributions

R.P. prepared all figures and lead the analysis, writing, and review process. R.P. and S.S. conducted the analyses and wrote the paper. S.S., E.M.F, E.D., and U.B. designed the simulation setup, U.B. performed the simulations, and S.D.H provided the land cover data. All co-authors provided feedback during the analysis and writing process.

## Competing interests

The authors declare no competing interests.
