## [Peer Review File · Nature Communications]

Global forestation and deforestation affect remote climate via adjusted atmosphere and ocean circulationREVIEWER COMMENTS

Reviewer #1 (Remarks to the Author):

Changes to the Earth's energy budget due to global forestation and deforestation affect remote climate via adjusted atmosphere and ocean circulation, by Portmann et al.

This study describes the results of idealized experiments with the Community Earth System Model (CESM) in which global land regions are changed to either be completely forested or completely deforested. It is found that this land cover perturbation has significant impacts on global mean temperature, ocean and atmospheric circulation and heat transports. While these experiments are idealized, I think they are useful to illustrate the global impacts that land cover perturbations can have and, given the push to increased forest cover to mitigate greenhouse gas impacts, it's worthwhile to motivate future studies that examine the impacts of this mitigation strategy to consider the fully coupled ocean-atmosphere-land system. I found the study to be well written and the results clearly presented. I only have minor comments to be considered before publication.

General comment:

The analysis mostly concentrates on annual averages, except for the Hadley cell part. Is there anything interesting to be mentioned about seasonality? e.g., perhaps changes to the monsoons? More substantial temperature impacts in the high latitudes during winter? I wouldn't suggest adding more extended figures as that can get frustrating to read, but if there is anything important, perhaps it could be mentioned in the text.

Comments by line number

I51: stating that reduction of tropical convection will "excite a poleward Rossby wave train" sounds strange to me. Really I think it's the convection that forces the wave train and reducing the tropical convection will be getting rid of that wave train. Suggest "...result in reduced tropical convection which will alter the tropical source of poleward propagating Rossby waves, thereby altering extratropical circulation and surface weather"

I57: it's unclear here what this "remarkable near-surface temperature response over the oceans" in the CMIP6 models is a response to. The CMIP6 simulations look at the response to many things. I assume

this is referring to some land-use change experiments that are part of LUMIP? I'd suggest just being clear about what this is a response to.

l86: I think it could be worth defining "heat days" in the text here. I realize they're defined in the caption but I don't think this is a commonly used term and it's brought in here a bit out of the blue.

l117: I think the reference to Fig 1b,c here should be for 1c,d?

l123: Similarly, I think the reference to Fig 1d,e should be to Fig 1e,f here.

l166: I find the sentence that begins "In combination..." to be confusing. This makes it sound like it is the increased sensible heat flux over land in combination with the weaker near surface wind speeds that is decreasing the sensible heat fluxes over ocean. I'm not sure if that's really what is meant but if so, it's not clear to me why increased sensible heat flux over land would lead to decreased sensible heat fluxes over oceans. Is it because the warmer air over land is being advected over ocean so there's not the same kind of contrast between the SST and the air temperature?

l167-169: I'm surprised that the role of the cooler North Atlantic in reducing the net energy fluxes into the atmosphere over the extratropical oceans isn't mentioned here. It's mentioned further down, but don't you think that would be playing a role here to reduce the sensible heat fluxes over the ocean?

l185: It's stated that "the warming of the forestation scenario and the cooling of the deforestation scenario in the extratropics is driven mainly by increased sensible heat fluxes". I have a couple of concerns about this statement. First, I think "increased sensible heat fluxes" can only refer to the forestation scenario, not the deforestation scenario. But the second is that I think the causal link here is a little misleading. In the forestation scenario, isn't the sensible heat flux increased primarily because the surface has warmed and ultimately that increased surface warming is due to reduced albedo and maybe also reduced latent heat fluxes. It doesn't sound right to invoke the sensible heat fluxes here as the ultimate driver.

l188: Instead of referring to Fig 6a here, it seems like Figs 6b and c are the relevant ones.

l195-201: Local arguments are invoked here for the changes in the stationary waves and I agree that this is likely an important contributor. But there could be a variety of non-local factors too. As mentioned earlier on, tropical convection has changed, which will change the Rossby wave source. The mean flow

has changed, which will alter the propagation and refraction of Rossby waves. I'm not sure that sufficient evidence has been provided to rule out the importance of these possibilities and leave a sole focus on the impacts of the forest cover on temperatures. Suggest either expanding on what the evidence is for this argument or include discussion of the other possibilities too.

l275: Perhaps remind readers at the beginning of this paragraph that you're referring to deforestation here. It has been a while since this was mentioned.

l306: "uttermost"  "utmost"

l335: A couple of things that I think could be expanded on in the methods description. (1) I don't think it's stated how long the standard preindustrial experiment is. All we know is that the first 200 years were omitted. Is it also 300 years long after year 200? If not and the control period that is being used does not correspond to the same time after initialization as the other experiments, it might be worth mentioning whether any drift in the simulations is substantial with the possibility of affecting the results. I don't think drift is much of a concern if your control period is also 300 years starting at year 200 and you're analyzing an equivalent time period. So if that's what is being done, then there's no need to mention the drift, but I think it would be good to clarify that this is what is being done.

l338: I think it could be worth making clear somewhere in this paragraph why some regions, such as the Sahara, don't experience much change. If I've understood correctly, it's only the portion of the grid cell that is not bare soil that is being changed to forest, so regions where there is a large fraction of bare ground will see less of an increase in the % area of forest. I realize this is may be obvious from the description here if you think about the method enough, but I think it could be worth making an explicit statement to this effect.

l375: I don't think the "Identification of jet streams" method section is quantitative enough. It would be very difficult for someone to reproduce the results as no information is given. What counts as "regions with enhanced wind speeds at the tropopause region"? How "pronounced" does the lower-tropospheric baroclinic zone have to be for it to count as a deep jet? Over what depths is baroclinicity confined in the upper troposphere for the shallow jets.

l396: "flxues"  "fluxes"

l398: In the [] that describes the sign convention I think it would be worth stating that this is referring to all fluxes. Right now it could be inferred that this is just referring to the snow melt at the surface.

l401: Maybe this is a standard terminology and if so you can ignore this comment. But it's confusing to me to refer to this as latent heat. I think I'd refer to it as latent energy. It's confusing to me because the $P \cdot \rho \cdot L$ term is actually "heating" the atmosphere. You've lost that latent energy when you precipitate the water, but you've gained heat. Maybe it's just me who finds this confusing...

General comments about figures:

(1) This is just a matter of preference, so feel free to take or leave. For figures, I prefer the method of stippling out regions that are insignificant as opposed to omitting them all together. In some cases its fine, but for example in Figures 2c, d, f, I feel like by not being able to see the global picture, you can't see the global patterns in which these significant regions are embedded. Even if regions are insignificant, they can still be helpful to see the global context.

(2) When you mention that only statistical significant results are shown, I think it would be worthwhile pointing toward the methods so readers know that that is where they can find the kind of significance testing that is being done.

(3) I think Figure 3 might be mentioned before Figure 2. At least most of the discussion surrounding Figure 3 happens first. If so, I'd suggest switching the order.

(4) In a number of the extended figures, you don't mention in the caption that the white regions are statistically insignificant.

Figure 2 caption: "arrows, shown where the differences are particularly large". How large? I don't see any reason to not be more specific here.

Extended Figure 2: I was curious why the clear sky albedo is changing so much over the ocean. Is this because of a change in water vapor? dust? optical properties of the ocean?

Extended Figure 5 caption. When the reference is made to panel (c), I think you mean (b)?

Extended Figure 6 caption. I think you need to make clear that the wind contours are the black ones and the temperature gradient contours are the green ones.

Reviewer #2 (Remarks to the Author):

This manuscript, “Changes to the Earth’s energy budget due to global forestation and deforestation affect remote climate via adjusted atmosphere and ocean circulation” by Portmann et al., addresses the question of how would substantial large-scale changes of forest cover impact the climate state with a state-of-the-art global climate model. Large-scale tree planting would have beneficial impact on carbon budget and ecosystem, but our highly-dimensional complex climate system requires examination of a wide spectrum of thermodynamic and dynamic responses to such envisioned land surface perturbations. The numerical experiments are framed in a simple manner, i.e., as “sledge-hammer” modifications to pre-industrial vegetation, but this is certainly an informative approach. On the side note: I would be interested to see these numerical experiments performed in the present climate and future projections, as well as using more gradual vegetation changes on decadal timescales, in a follow-up study.

I would like to see this manuscript published in Nature Communications after the authors address the following minor points.

Since changes in AMOC are so important in CESM2 under these explored preindustrial changes, I would like to see Figure 4 (AMOC changes) also in an isopycnal framework (e.g., using simply Sigma_2 as the vertical coordinate, or the authors could opt for a more elaborate approach by using neutral density – check for example TEOS-10 manual) along with a succinct discussion which should involve analysis of the changes in the associated water mass transformations in the Atlantic.

Furthermore, I would be curious to see analysis of the changes in the ocean heat content (e.g., down to 700m, down to 2000m, and between 700m and 2000m).

Line 333: Please remind audience her of how many vertical levels are in the atmosphere and the oceans in this version of CEMS2

Reviewer #3 (Remarks to the Author):

In this study climate responses to global forestation and deforestation are quantified from the coupled Community Earth System Model (CESM) preindustrial control and sensitivity experiments, with a focus on the changes in the Earth's energy balance and atmospheric and oceanic circulations. There are significantly interesting and noteworthy results in this manuscript, compared to previous studies that focused on the thermodynamic effects of land surface changes in the literature, but I have several major concerns on the experiment design and analysis.

1) The global forestation and deforestation are very idealized scenarios. Preindustrial conditions were used as the base-state climate forcing. Any potential interactions and/or feedbacks from carbon dioxide, short-lived climate forcers, and anthropogenic land use and land cover changes are not considered. Thus, the quantified radiative forcing, global temperature, snow/ice albedo, and circulation changes, including the AMOC strength change, are not exactly comparable to previous studies on the historical transient and projected future climate changes that are cited at various places. The contrast between forestation and deforestation could be quite different under the present-day conditions. Please provide a justification and discussion on these perspectives.

2) The control experiment with preindustrial climate forcing was first conducted for 200 years, and then the two sensitivity experiments were branched and run for 300 years. According to the time series of some key variables in Extended Fig 5, the extended control experiment appears to be in equilibrium, but climate responses beyond decadal timescales in both sensitivity experiments are not equilibrated yet, indicating that the CESM simulations was not fully spun up for the given global forestation and deforestation perturbations. How do the time varying responses compare to similar CESM CMIP5/6 simulations? Please discuss on this.

3) The experiments are based on a single model simulation (without ensemble members). It's unclear how the statistical significance and robustness are characterized for all the spatial distribution plots. For example, the panel a-d of Fig 1. This needs to be justified according to the known behaviors from existing CESM Large Ensembles.

4) The analysis of changes in energy balance, albedo and precipitation is rather shallow. More in-depth analysis on the thermodynamic changes (e.g., changes in convection and clouds) and seasonal monsoon precipitation to put this study in the context of the established literature.

Global forestation and deforestation affect remote climate via adjusted atmosphere and ocean circulation

REPLIES TO THE REVIEWERS' COMMENTS

Raphael Portmann, Urs Beyerle, Edouard Davin, Erich M. Fischer, Steven De Hertog, Sebastian Schemm

We thank the reviewers for their helpful comments. Below, we address all the points raised (in blue). We provide some additional analyses and diagrams in this document and in new or updated figures in the manuscript. Comments from all reviewers as well as some adjustments to comply with the Nature Communications formatting guidelines have resulted in the following general changes to the manuscript.

- We now explain the motivation for, and rationale of the study design in more detail in the first section and the methods section and discuss its implications in the conclusions (in response to Reviewer #3).
- Instead of extended Figures, we now provide a separate “Supplementary Information” document with Supplementary Figures and methods
- We now discuss relevant seasonal aspects of the most important simulated changes and provide the corresponding Figures in the Supplementary Information (in response to Reviewer #1).
- We provide an additional analysis of the Rossby wave source as Supplementary Figure
- We add two panels in Figure 1 to show and discuss changes in cloud cover
- We extend the analysis of the ocean part resulting in two additional Figures which are provided in the Supplementary Information (in response to Reviewer #2).
- We shortened the abstract and added a new paragraph at the end of the introduction that summarizes the key results and conclusions.
- We shortened the title.

REVIEWER COMMENTS

Reviewer #1 (Remarks to the Author):

Changes to the Earth's energy budget due to global forestation and deforestation affect remote climate via adjusted atmosphere and ocean circulation, by Portmann et al.

This study describes the results of idealized experiments with the Community Earth System Model (CESM) in which global land regions are changed to either be completely forested or completely deforested. It is found that this land cover perturbation has significant impacts on global mean temperature, ocean and atmospheric circulation and heat transports. While these experiments are idealized, I think they are useful to illustrate the global impacts that land cover perturbations can have and, given the push to increased forest cover to mitigate greenhouse gas impacts, it's worthwhile to motivate future studies that examine the impacts of this mitigation strategy to consider the fully coupled ocean-atmosphere-land system. I found the study to be well written and the results clearly presented. I only have minor comments to be considered before publication.

General comment:

The analysis mostly concentrates on annual averages, except for the Hadley cell part. Is there anything interesting to be mentioned about seasonality? e.g., perhaps changes to the monsoons? More substantial temperature impacts in the high latitudes during winter? I wouldn't suggest adding more extended figures as that can get frustrating to read, but if there is anything important, perhaps it could be mentioned in the text.

Good suggestion. Reviewer 3 had similar suggestions. We now highlight certain aspects of seasonality in the text. For more details, we refer to the response to the last comment of reviewer 3.

Comments by line number

l51: stating that reduction of tropical convection will "excite a poleward Rossby wave train" sounds strange to me. Really I think it's the convection that forces the wave train and reducing the tropical convection will be getting rid of that wave train. Suggest "...result in reduced tropical convection which will alter the tropical source of poleward propagating Rossby waves, thereby altering extratropical circulation and surface weather"

Thanks, we agree and changed the text accordingly, it now reads:

Complete deforestation of the tropics may result in reduced tropical convection which **-weakens the tropical source of poleward propagating Rossby waves, thereby altering** extratropical circulation and surface weather [18,19,20].

I57: it's unclear here what this "remarkable near-surface temperature response over the oceans" in the CMIP6 models is a response to. The CMIP6 simulations look at the response to many things. I assume this is referring to some land-use change experiments that are part of LUMIP? I'd suggest just being clear about what this is a response to.

Agreed. We clarified this aspect. We now state:

Recent efforts based on the Land-Use Model Intercomparison Project (LUMIP) contribution to the Coupled Model Intercomparison Project Phase 6 (CMIP6) showed that several models respond to global deforestation with remarkable near-surface temperature changes over the oceans [10]

I86: I think it could be worth defining "heat days" in the text here. I realize they're defined in the caption but I don't think this is a commonly used term and it's brought in here a bit out of the blue.

Agreed, we added the definition in the text.

Surface warming leads to more heat days, which are defined as days with maximum temperature above 30°C.

I117: I think the reference to Fig 1b,c here should be for 1c,d?

Thanks for spotting this. We corrected it.

I123: Similarly, I think the reference to Fig 1d,e should be to Fig 1e,f here.

Thanks again, corrected.

I166: I find the sentence that begins "In combination..." to be confusing. This makes it sound like it is the increased sensible heat flux over land in combination with the weaker near surface wind speeds that is decreasing the sensible heat fluxes over ocean. I'm not sure if that's really what is meant but if so, it's not clear to me why increased sensible heat flux over land would lead to decreased sensible heat fluxes over oceans. Is it because the warmer air over land is being advected over ocean so there's not the same kind of contrast between the SST and the air temperature?

Thanks for pointing out this aspect. We clarified it in the text.

When this warmer air is advected over the adjacent ocean, sensible heat fluxes over the ocean decrease due to a reduced air-sea temperature contrast (see Extended Fig. 8b). This is further supported by the weaker near-surface winds, which also contribute to reduced ocean-atmosphere latent heat fluxes. At the same time, longwave cooling of the atmosphere over the extratropical oceans is enhanced due to warmer air temperatures.

I167-169: I'm surprised that the role of the cooler North Atlantic in reducing the net energy fluxes into the atmosphere over the extratropical oceans isn't mentioned here. It's mentioned further down, but don't you think that would be playing a role here to reduce the sensible heat fluxes over the ocean?

Thanks for mentioning this point. Note that this part of the paragraph talks about the first 40 years in the simulation. In fact, during the first 40 years, zonal mean SSTs are not reduced. This is mainly because the strong cooling over the North Atlantic starts only after this 40-year period, due to the delayed response of the ocean. We have made it clearer in the text that we are only referring to the first 40 years at this point.

In total, the extratropical atmosphere between 40 and 70°N receives more additional energy over land than it loses over the oceans **during the first four decades of the simulation.**

l185: It's stated that "the warming of the forestation scenario and the cooling of the deforestation scenario in the extratropics is driven mainly by increased sensible heat fluxes". I have a couple of concerns about this statement. First, I think "increased sensible heat fluxes" can only refer to the forestation scenario, not the deforestation scenario. But the second is that I think the causal link here is a little misleading. In the forestation scenario, isn't the sensible heat flux increased primarily because the surface has warmed and ultimately that increased surface warming is due to reduced albedo and maybe also reduced latent heat fluxes. It doesn't sound right to invoke the sensible heat fluxes here as the ultimate driver.

Thank you for pointing out this inconsistency. We agree that this statement only applies to the afforestation scenario. Regarding your second concern, we also fully agree that the key reason for the warming (cooling) is the decrease (increase) in albedo and the resulting changes in surface temperature. The causes of the warming are extensively discussed the first results section. We have consequently deleted the subordinate clause. The paragraph now reads

The warming in the forestation scenario and the cooling in the deforestation scenario in the extratropics extend **through** the entire depth of the troposphere and **temperature changes are largest** in middle latitudes in *forest* and **in middle and** polar latitudes in *grass*.

l188: Instead of referring to Fig 6a here, it seems like Figs 6b and c are the relevant ones. Agreed and changed accordingly.

l195-201: Local arguments are invoked here for the changes in the stationary waves and I agree that this is likely an important contributor. But there could be a variety of non-local factors too. As mentioned earlier on, tropical convection has changed, which will change the Rossby wave source. The mean flow has changed, which will alter the propagation and refraction of Rossby waves. I'm not sure that sufficient evidence has been provided to rule out the importance of these possibilities and leave a sole focus on the impacts of the forest cover on temperatures. Suggest either expanding on what the evidence is for this argument or include discussion of the other possibilities too.

We agree with you that changes in tropical Rossby wave sources and their interaction with a changed midlatitude background flow act as additional non-local drivers behind the changes in the stationary Rossby wave pattern. To give an impression of this change, we computed the Rossby wave source following Sardeshmukh and Hoskins (1988) and show the difference between the two scenarios below. The control run reveals Rossby wave sources that are largely consistent with Nie et al (2019), who based their computation on ERA-interim. The differences between the perturbed simulations and the control

run don't show pronounced changes in the tropics, which is why changes in tropical RWS are unlikely drivers of the mid-latitude geopotential height difference found between the two perturbed simulations and the control run.

We added an additional statement to the manuscript which reads as follows:

Additional modification of stationary waves may arise from changes in tropical Rossby wave sources. A closer inspection of changes in tropical Rossby wave sources [Sardeshmukh and Hoskins, 1988] however reveals no well-marked changes between the two scenarios and the control experiment (Supplementary Fig. 14)

Figure 1: Climatological mean Rossby wave source following Sardeshmukh and Hoskins (1988) at 300 hPa for (a) control, (b) the difference between forest and control, and (c) the difference between grass and control. There are no well-marked changes in tropical latitudes.

I275: Perhaps remind readers at the beginning of this paragraph that you're referring to deforestation here. It has been a while since this was mentioned.

Thanks for this suggestion, we now state

In boreal summer, the Southern Hemispheric branch of the Hadley cell weakens in the deforestation simulation (Fig. 5d) as a result of the reduced inter-hemispheric temperature contrast

I306: "uttermost"  "utmost"

Changed as suggested.

I335: A couple of things that I think could be expanded on in the methods description. (1) I don't think it's stated how long the standard preindustrial experiment is. All we know is that the first 200 years were omitted. Is it also 300 years long after year 200? If not and the control period that is being used does not correspond to the same time after initialization as the other experiments, it might be worth mentioning whether any drift in the simulations is substantial with the possibility of affecting the results. I don't think drift is much of a concern if your control period is also 300 years starting at year 200 and you're analyzing an equivalent time period. So if that's what is being done, then there's no need to mention the drift, but I think it would be good to clarify that this is what is being done.

Thanks. This is what is being done. We clarified this aspect, we now state

After these 200 years the forestation and deforestation experiments as well as the control simulation were branched off and run for 300 years.

I338: I think it could be worth making clear somewhere in this paragraph why some regions, such as the Sahara, don't experience much change. If I've understood correctly, it's only the portion of the grid cell that is not bare soil that is being changed to forest, so regions where there is a large fraction of bare ground will see less of an increase in the % area of forest. I realize this is may be obvious from the description here if you think about the method enough, but I think it could be worth making an explicit statement to this effect.

We added a sentence to clarify this aspect in the methods section, which reads

As a consequence of this approach, regions with large fractions of bare soil (e.g. deserts) do not exhibit large changes.

I375: I don't think the "Identification of jet streams" method section is quantitative enough. It would be very difficult for someone to reproduce the results as no information is given. What counts as "regions with enhanced wind speeds at the tropopause region"? How "pronounced" does the lower-tropospheric baroclinic zone have to be for it to count as a deep jet? Over what depths is baroclinicity confined in the upper troposphere for the shallow jets.

Thanks for pointing this out, we only referenced the corresponding paper that introduced the method (Koch et al. 2006). We extended this section, which now describes the jet identification method more quantitatively. Changes are shown below:

Jet streams are identified as instantaneous regions with enhanced wind speeds in the tropopause region [38] based on 6-hourly model output. This method is based on the vertical average of the total wind speed between 400 and 100 hPa defined as

$$\alpha_{vel} = \frac{1}{400 \text{ hPa} - 100 \text{ hPa}} \int_{100 \text{ hPa}}^{400 \text{ hPa}} (u^2 + v^2)^{\frac{1}{2}} dp \quad (3)$$

where u and v are the zonal and meridional wind components respectively. Jet streams are then identified as regions with $\alpha_{vel} \geq 30 \text{ ms}^{-1}$. Further, the jet streams are separated into deep and shallow jets based on an index of upper-tropospheric wind shear

$$\Delta v_{rel} = \frac{v_{200} - v_{500}}{v_{200}} \quad (4)$$

where v_{200} and v_{500} denote the horizontal wind speeds at 200 hPa and 500 hPa respectively. Deep jets ~~are required to be located above a pronounced~~ exhibit low-upper tropospheric wind shear ($\Delta v_{rel} < 0.4$) and hence are accompanied predominantly by lower-tropospheric ~~baroclinic zone and are~~ baroclinicity. They can therefore be considered as manifestations of the extratropical or eddy-driven jet stream. For shallow jets, baroclinicity is confined to the upper troposphere ($\Delta v_{rel} > 0.4$) and they are considered manifestations of the subtropical or thermally-driven jet stream. For the more detailed rationale behind this method and the selection of the thresholds for α_{vel} and Δv_{rel} the reader is referred to [38].

l396: "flxues"  "fluxes"

Thanks for spotting; changed.

l398: In the [] that describes the sign convention I think it would be worth stating that this is referring to all fluxes. Right now it could be inferred that this is just referring to the snow melt at the surface.

Good suggestion, we changed the note in [] to clarify that it refers to all fluxes. It now reads

[here: all fluxes are positive if net energy input is into the atmosphere]

l401: Maybe this is a standard terminology and if so you can ignore this comment. But it's confusing to me to refer to this as latent heat. I think I'd refer to it as latent energy. It's confusing to me because the $P \cdot \rho \cdot L$ term is actually "heating" the atmosphere. You've lost that latent energy when you precipitate the water, but you've gained heat. Maybe it's just me who finds this confusing...

In our understanding, latent heat and latent energy are largely synonyms. In this case, we see latent heat as energy stored in the water vapor that is released as sensible heat when the phase change to liquid water occurs. Hence, condensation is basically the transformation of latent heat/latent energy to sensible heat. It seems standard terminology to us, so we decided to leave it as it is.

General comments about figures:

(1) This is just a matter of preference, so feel free to take or leave. For figures, I prefer the method of stippling out regions that are insignificant as opposed to omitting them all together. In some cases its fine, but for example in Figures 2c, d, f, I feel like by not being able to see the global picture, you can't see the global patterns in which these significant regions are embedded. Even if regions are insignificant, they can still be helpful to see the global context.

Thanks for this excellent suggestion. We now hatch the insignificant regions in all figures where it provides more clarity. Please note that in Figure 2c/e (or Figure 4c/e in the revised manuscript) changes are significant practically everywhere and the white regions just exhibit comparatively small changes in geopotential height (we adjusted the colorbar to include the white regions to clarify this). Further, in Figures 2d/f we unmask regions with jet frequencies below 10% and adjusted the colormap to show a global picture.

(2) When you mention that only statistical significant results are shown, I think it would be worthwhile pointing toward the methods so readers know that that is where they can find the kind of significance testing that is being done.

Thanks for this suggestion. The testing method is a two-sided Wilcoxon rank-sum test applied to annual mean fields, subsequently corrected with a Benjamini-Hochberg correction to control the false discovery rate at a level of 0.1. This procedure is described in the methods section. We now refer to the methods section in the Figure captions, for example by stating "Statistically insignificant differences (see methods) are hatched."

(3) I think Figure 3 might be mentioned before Figure 2. At least most of the discussion surrounding Figure 3 happens first. If so, I'd suggest switching the order.

Thanks for noticing. We moved Figure 3 and 4 before Figure 2 to have the correct order.

(4) In a number of the extended figures, you don't mention in the caption that the white regions are statistically insignificant.

Thanks for pointing this out. We now explain in every caption which regions are / are not statistically significant.

Figure 2 caption: "arrows, shown where the differences are particularly large". How large? I don't see any reason to not be more specific here.

Good point. We tried to circumvent a specific description because it is, for cosmetic reasons, not very simple. In the revised version we are now more specific and clearly state which arrows are shown. We state

arrows, only the sixty globally largest differences at every tenth latitude and longitude index are shown.

Extended Figure 2: I was curious why the clear sky albedo is changing so much over the ocean. Is this because of a change in water vapor? dust? optical properties of the ocean?

The large albedo changes over high latitude oceans are directly linked to changes in sea-ice cover. However, there is likely no simple explanation for the relatively smaller changes over the rest of the oceans.

Extended Figure 5 caption. When the reference is made to panel (c), I think you mean (b)?

Thanks for spotting this one. We corrected it.

Extended Figure 6 caption. I think you need to make clear that the wind contours are the black ones and the temperature gradient contours are the green ones.

Thanks for this suggestion, we clarified.

Reviewer #2 (Remarks to the Author):

This manuscript, “Changes to the Earth’s energy budget due to global forestation and deforestation affect remote climate via adjusted atmosphere and ocean circulation” by Portmann et al., addresses the question of how would substantial large-scale changes of forest cover impact the climate state with a state-of-the-art global climate model. Large-scale tree planting would have beneficial impact on carbon budget and ecosystem, but our highly-dimensional complex climate system requires examination of a wide spectrum of thermodynamic and dynamic responses to such envisioned land surface perturbations. The numerical experiments are framed in a simple manner, i.e., as “sledge-hammer” modifications to pre-industrial vegetation, but this is certainly an informative approach. On the side note: I would be interested to see these numerical experiments performed in the present climate and future projections, as well as using more gradual vegetation changes on decadal timescales, in a follow-up study.

I would like to see this manuscript published in Nature Communications after the authors address the following minor points.

1) Since changes in AMOC are so important in CESM2 under these explored preindustrial changes, I would like to see Figure 4 (AMOC changes) also in an isopycnal framework (e.g., using simply Sigma_2 as the vertical coordinate, or the authors could opt for a more elaborate approach by using neutral density – check for example TEOS-10 manual) along with a succinct discussion which should involve analysis of the changes in the associated water mass transformations in the Atlantic.

Thanks for this suggestion and your positive evaluation of our manuscript. We interpolated the data to an isopycnal coordinate system (sigma_2) and computed AMOC and its changes in this new coordinate system based on an open-source python package¹ (see Figure 2 below).

In the isopycnal coordinate system, the AMOC maximum is located further poleward than in the z-coordinate system (i.e. closer to where deep convection occurs). Consequently, also the strongest AMOC changes occur further poleward than in height-coordinates. In our opinion, these Figures show in essence the same key message as the ones in Figure 4 in the manuscript but are now closer to the source location of deep convection. The overturning circulation is enhanced in grass and weakened in forest. The isopycnal representation appears to better emphasize changes in water mass transformations, i.e. deep ocean convection.

We added the following three sentences in the discussion and provide the Figure below in the Supplementary Material.

¹ Stephen G Yeager, POP_MOC,
https://github.com/sgyeager/POP_MOC/blob/main/notebooks/pop_MOCsig2_1deg.ipynb

To diagnose changes in water mass transformation, AMOC is also computed in potential density coordinates (Supplementary Fig. 8). In this framework, the AMOC maximum and also the strongest simulated AMOC changes occur further north (between 40-60°N) than in height coordinates. This points at the key role of changes in water mass transformations in the subpolar North Atlantic for the changes in AMOC strength.

Figure 2: Annual mean meridional mass stream function in the Atlantic in isopycnal framework (potential density anomaly as vertical coordinate), i.e. the AMOC (contours, in Sv, only values above 4 Sv shown with a 4 Sv interval) for (a) forest (black) and control (gray) and (b) grass (black) and control (gray), as well as differences of experiments relative to control (shading, in Sv). Statistically insignificant differences are shown in white.

2) Furthermore, I would be curious to see analysis of the changes in the ocean heat content (e.g., down to 700m, down to 2000m, and between 700m and 2000m).

We have extended the ocean analysis to reveal changes in heat content in different layers. It clearly shows that the Atlantic Ocean is the global hot spot of heat content changes. In the forest simulation, the heat content in the mid- and upper Atlantic Ocean is increased and in the deep ocean it is decreased. This agrees with reduced deep overturning. In grass, a roughly reversed pattern is visible. For this experiment, large losses in upper ocean heat content occur also in the Pacific which are probably related to increased heat loss to the atmosphere. Overall, this additional supplementary Figure puts further emphasis on the important role of the (North-) Atlantic Ocean response to global-scale land use change as driver behind the adjusted energy transport in the Earth System.

We added Figure 3, which is shown below, to the Supplementary information of the paper. We also added a brief discussion in the manuscript, which reads:

The critical role that the oceans, and in particular the Atlantic Ocean, play in the climate systems response to forestation and deforestation is also supported by the substantial increase in ocean heat content in *forest* and its decrease in *grass* in the mid- and upper oceans (with the exception of the North Atlantic warming hole region, see Supplementary Fig. 12). In the forestation simulation, the heat content decreases in the deep ocean, likely due to the shallower and weaker AMOC, which transports heat less efficiently to the deep ocean.

Figure 3: Absolute changes of the ocean heat content (shading, in Jm^{-2} from (a,b) 0-708 m (c,d) 708-1970 and (e,f) 1970 m to ocean bottom in (a,c,e) forest and (b,d,f) grass. Please ignore artefacts at the ocean boundaries.

3) Line 333: Please remind audience her of how many vertical levels are in the atmosphere and the oceans in this version of CEMS2

Thanks for this suggestion, we now mention the number of levels in the atmosphere model (CAM6, 32 vertical levels) and ocean model (POP2, 60 vertical levels).

Reviewer #3:

In this study climate responses to global forestation and deforestation are quantified from the coupled Community Earth System Model (CESM) preindustrial control and sensitivity experiments, with a focus on the changes in the Earth's energy balance and atmospheric and oceanic circulations. There are significantly interesting and noteworthy results in this manuscript, compared to previous studies that focused on the thermodynamic effects of land surface changes in the literature, but I have several major concerns on the experiment design and analysis.

1) The global forestation and deforestation are very idealized scenarios. Preindustrial conditions were used as the base-state climate forcing. Any potential interactions and/or feedbacks from carbon dioxide, short-lived climate forcers, and anthropogenic land use and land cover changes are not considered. Thus, the quantified radiative forcing, global temperature, snow/ice albedo, and circulation changes, including the AMOC strength change, are not exactly comparable to previous studies on the historical transient and projected future climate changes that are cited at various places. The contrast between forestation and deforestation could be quite different under the present-day conditions. Please provide a justification and discussion on these perspectives.

Thank you for this comment. We fully agree that the results from this idealized approach are not readily comparable to transient historical or future simulations related to land use changes. The advantage of our approach is that we exclude potential confounding factors (i.e., changes in response to historical greenhouse gas, aerosol, and natural forcing) that may affect transient simulations and would be challenging to disentangle, given the high internal variability.

In a next step similar simulations are planned under more realistic (future) scenarios. For example, we show that the strong cooling in our deforestation scenario is largely due to strong snow-albedo feedbacks over Northern high-latitude land. Hence, under warmer climate conditions with lower snow cover over North America and Eurasia, the responses in circulation patterns would also differ. We see our study as a first step toward isolating the response in an idealized study in which we lay the foundations for such further studies by pointing at the basic mechanisms how large-scale land use changes affect the atmospheric and ocean circulation. We added some more discussion in the first paragraph of the first section and the conclusions, which read as follows:

First section

Our goal is to understand the fundamental effects that forestation and deforestation have on the atmospheric and ocean circulation. Therefore, this idealized study is a first step towards assessing the impacts of more realistic forestation and deforestation scenarios on circulation in past, present, and future climates.

Conclusions

As a next step, research is needed to assess the mechanisms described here under more complex model configurations, e.g. considering different scales of forestation [Lague et al, 2016], focusing on individual regions [Gedney and Valdes, 2000], including transient future climate scenarios [Koch et al. 2021] and adding biogeochemical effects [Boysen et al. 2020].

2) The control experiment with preindustrial climate forcing was first conducted for 200 years, and then the two sensitivity experiments were branched and run for 300 years. According to the time series of some key variables in Extended Fig 5, the extended control experiment appears to be in equilibrium, but climate responses beyond decadal timescales in both sensitivity experiments are not equilibrated yet, indicating that the CESM simulations was not fully spun up for the given global forestation and deforestation perturbations. How do the time varying responses compare to similar CESM CMIP5/6 simulations? Please discuss on this.

Thanks for raising this important point. Indeed, the control run is clearly in equilibrium. After the land use perturbations, both simulations show a relatively rapid initial adjustment of global mean temperatures within the first five decades of the simulations. Therefore, we exclude the first 50 years from the computation of climatological means. You are right that even after this first adjustment, the two experiments are not yet in equilibrium. However, even if we exclude the first 100 years from the analysis, the results remain qualitatively the same. As shown by our radiative forcing calculations, the simulations are not even fully equilibrated after 300 years. This is in line with long time scales of adjustments of the deep ocean to global scale land use change (see e.g. Renssen et al 2003). However, we assume that they are approaching a new equilibrium and our analysis provides a conservative estimate of the response to the forestation changes. Further, current scientific interest around climate mitigation focuses mainly on the coming century which is why it seems reasonable to focus on roughly this time scale. The 50 years allowed for spin-up in our study is comparable what has been used for other similar setups. In previous global deforestation experiments within the framework of CMIP6 (Lawrence et al 2016, Boysen et al. 2020) the first 30 years are omitted after a linear forest removal over 50 years. We expanded on this in the methods section as follows:

The five-decade long spin-up period is justified by the observation that this is approximately the period during which the initial rapid adjustment of parameters such as the 2-metre temperature, albedo and snow/sea ice fractions takes place (see Fig. 5). It is important to note that both experiments are not yet in full equilibrium after these five decades and even not after three centuries, which can be inferred from radiative forcing estimation. Excluding the first 100 years from the analysis results in qualitatively similar results. A spin-up period of five decades for the initial adjustment period is in line with previous CMIP6 experiments on global deforestation, which omit the first three decades after a linear deforestation over 50 years (e.g. Lawrence et al. 2016, Boysen et al. 2020)

3) The experiments are based on a single model simulation (without ensemble members). It's unclear how the statistical significance and robustness are characterized for all the spatial distribution plots. For example, the panel a-d of Fig 1. This needs to be justified according to the known behaviors from existing CESM Large Ensembles.

Thanks for this comment. Internal variability can be sampled by either running large initial condition ensembles or longer simulations. Given that these simulations take time to equilibrate and that all forcings are fixed we here produce one multi-century run per experiment instead of several shorter initial condition members. Multi-century simulations allow for a robust statistical assessment of differences between annual means with the test procedure discussed in the methods section.

We added an explicit reference to the methods section, which describes the statistical significance testing, in the figure captions. Also, we added a sentence in the methods section that hopefully helps to clarify the procedure used here. The sentence reads

The test data for each experiment consist of annual or seasonal means values of each year from 50 to 300.

4) The analysis of changes in energy balance, albedo and precipitation is rather shallow. More in-depth analysis on the thermodynamic changes (e.g., changes in convection and clouds) and seasonal monsoon precipitation to put this study in the context of the established literature.

Thank you for this suggestion. Reviewer 1 made similar comments related to the seasonality. We now discuss seasonal changes as well. However, length constraints limit the seasonal analysis to circulation and clouds changes, which have, in contrast to thermodynamic changes (Laguë et al. 2021) received less attention in previous studies. We believe that the presented analyses of albedo and energy balance changes are already sufficiently deep to provide the necessary basis for the analysis of dynamic changes.

Overall, our manuscript benefits from a more detailed seasonal discussion of changes in circulation and clouds. Specifically, we included another row in Figure 1 of the manuscript showing the changes in total cloud cover as requested and briefly discuss their seasonality. For your reference, the seasonal plots for precipitation, temperature, and clouds are shown in Figures 4 and 5, and the seasonal plots for jet streams are in Figures 6 and 7, along with the additional sentences added to the manuscript are also shown below. The Figures are also added to the supplementary information of the paper.

(...)

For both simulations, temperature changes over northern extratropical land are larger in boreal summer (April to September) compared to winter (October to March) (Supplementary Fig. 5c,d and 6c,d). For grass, this is consistent with the albedo effect due to late spring/summertime high latitude snow cover. For forest however, this remains less clear and is possibly linked to reduced low-level cloud cover in summer (Supplementary Fig. 5e).

(...)

Global mean cloud cover decreases in forest and increases in grass (Fig. 1e,f). Over northern extratropical land cloud cover is reduced by more than 2.5 % in forest and increased by up to 10 % grass and changes are more pronounced in boreal summer compared to winter (for seasonal changes see Supplementary Fig. 5e,f and 6e,f). This partly contradicts recent observational evidence [Duveiller et al. 2021, Xu et al. 2022] but is in agreement with [Lague et al. 2016] who argue that, after a certain degree of mid-latitude forestation, increased sensible heat fluxes lead to warming and drying of the troposphere and thereby inhibit cloud formation. Over tropical land, changes in cloud cover are less pronounced and their sign depends on the region. Cloud cover changes also occur over oceans, pointing towards changes of the ocean circulation and ocean-atmosphere interactions but also of the mid-latitude storm tracks. In forest, cloud cover increases particularly strongly over the eastern North Atlantic [up to 7.5 %, mostly in boreal summer] and the tropical South Pacific [up to 5 %] (Fig. 1f). In grass, cloud cover increases particularly strongly over the eastern North Pacific and the western North

Atlantic (5-10 %) and decreases over a relatively confined region over the eastern tropical South Pacific (up to 10 %). In many regions with large changes, low-clouds contribute strongly to the response.

(...)

Precipitation decreases in the Euro-Mediterranean region in forest and an increases in grass by 10 % and more. In both simulations, relative changes are larger in boreal summer than in winter (Supplementary Fig. 5g,h and 6g,h). Tropical precipitation changes are characterized by latitudinal shifts which depend remarkably little on the seasons. Exceptions with pronounced seasonal differences are the equatorial North Atlantic/West Africa (much larger relative increase in boreal winter) in forest and the western tropical Pacific (stronger decrease in boreal summer) and East Africa (changes occur mostly in boreal summer) in grass.

(...)

There is a pronounced seasonality in the response of the extratropical jet stream in forest, particularly in the Northern Hemisphere (Supplementary Fig. 16c and 17c). Absolute changes are larger in boreal winter than in boreal summer. Further, the North Pacific jet stream predominantly shifts poleward in boreal winter and weakens in boreal summer. In the deforestation scenario, the changes in the mean midlatitude circulation patterns are generally reversed and stronger than those found in the forestation scenario including enhanced eddy kinetic energy (Supplementary Fig. 15 b), stronger westerlies (Fig. 3e), stronger eddy momentum flux convergence and more frequent deep jets (Fig. 3f), but without latitudinal shift of the jet stream in the Northern Hemisphere. The jet intensifies primarily over the western North Atlantic in boreal winter and across the whole northern mid-latitudes in boreal summer (Supplementary Fig. 16e and 17e).

(...)

The regional decrease in sea surface temperatures related to the slow-down in the Atlantic meridional overturning circulation (Fig. 4) drive a local atmospheric temperature and circulation anomaly, which enables enhanced baroclinic growth of synoptic-scale eddies and subsequently eddy kinetic energy (Extended Fig. 11a) and eddy-momentum flux convergence downstream that ultimately shifts the eddy-driven jet over the northeastern North Atlantic poleward (Fig. 3d) [49] despite the hemispheric-wide decrease in near-surface baroclinicity. This effect is more pronounced in boreal winter but occurs also in boreal summer.

Figure 4: Seasonal mean differences between forest and control for (a,b) precipitation (relative differences), (c,d) 2 m temperature, and (e,f) total cloud fraction. Shown are means from (left column) October to March and (right column) April to September. Statistically insignificant differences are hatched (see methods section)

Figure 5: Seasonal mean differences between grass and control for (a,b) precipitation (relative differences), (c,d) 2 m temperature, and (e,f) total cloud fraction. Shown are means from (left column) October to March and (right column) April to September. Statistically insignificant differences are hatched (see methods section)

Figure 6: Mean October to March frequencies of (a,c,e) deep jets and (b,d,f) shallow jets in (a,b) control (shading, in %) and (c,d) differences in forest and (e,f) differences in grass relative to control. Statistically insignificant differences are hatched (see methods section)

Figure 7: Mean April to September frequencies of (a,c,e) deep jets and (b,d,f) shallow jets in (a,b) control (shading, in %) and (c,d) differences in forest and (e,f) differences in grass relative to control. Statistically insignificant differences are hatched (see methods section)

References

- Boysen, L. R., Brovkin, V., Pongratz, J., Lawrence, D. M., Lawrence, P., Vuichard, N., Peylin, P., Liddicoat, S., Hajima, T., Zhang, Y., Rocher, M., Delire, C., Séférian, R., Arora, V. K., Nieradzik, L., Anthoni, P., Thiery, W., Laguë, M. M., Lawrence, D., and Lo, M.-H. (2020): Global climate response to idealized deforestation in CMIP6 models, *Biogeosciences*, 17, 5615–5638, <https://doi.org/10.5194/bg-17-5615-2020>,
- Laguë, M. M., Swann, A. L. S., & Boos, W. R. (2021). Radiative Feedbacks on Land Surface Change and Associated Tropical Precipitation Shifts, *Journal of Climate*, 34(16), 6651-6672, <https://journals.ametsoc.org/view/journals/clim/34/16/JCLI-D-20-0883.1.xml>
- Lawrence, D. M., Hurtt, G. C., Arneeth, A., Brovkin, V., Calvin, K. V., Jones, A. D., Jones, C. D., Lawrence, P. J., de Noblet-Ducoudré, N., Pongratz, J., Seneviratne, S. I., and Shevliakova, E. (2016) : The Land Use Model Intercomparison Project (LUMIP) contribution to CMIP6: rationale and experimental design, *Geosci. Model Dev.*, 9, 2973–2998, <https://doi.org/10.5194/gmd-9-2973-2016>,
- Nie, Y., Zhang, Y., Yang, X.-Q., & Ren, H.-L. (2019). Winter and summer Rossby wave sources in the CMIP5 models. *Earth and Space Science*, 6, 1831– 1846. <https://doi.org/10.1029/2019EA000674>
- Renssen, H., Goosse, H., and Fichefet, T. (2003), On the non-linear response of the ocean thermohaline circulation to global deforestation, *Geophys. Res. Lett.*, 30, 1061, doi:10.1029/2002GL016155, 2.
- Sardeshmukh, P. D., & Hoskins, B. J. (1988). The Generation of Global Rotational Flow by Steady Idealized Tropical Divergence, *Journal of Atmospheric Sciences*, 45(7), 1228-1251. https://journals.ametsoc.org/view/journals/atsc/45/7/1520-0469_1988_045_1228_tgogrf_2_0_co_2.xml

[revised manuscript text omitted]
 non-linear color spacing) compared to *control* for (a,c,e) *forest* and (b,d,f) *grass*. Only statistically significant results are shown. Statistically insignificant results are shown hatched, see methods section.

Figure 2: Annual-mean meridional heat transport for Changes of the annual mean meridional heat transport in atmosphere and ocean. Shown are (a) meridional heat transport in control and the differences between control and the two experiments, (b) forest-control its changes in forest with respect to control, and (c) grass-control its changes in grass with respect to control. The total heat transport (black) is separated into atmospheric (red) and ocean (blue) transport. Atmospheric transport is further separated into the transport of moist (dotted) and dry static energy (dashed).

Figure 3: Annual-Slow-down and acceleration of the Atlantic meridional overturning circulation. Shown is the annual mean meridional mass stream function in the Atlantic, i.e. the AMOC (contours, in Sv, only values above 4 Sv shown with a 4 Sv interval) for (a) forest (black) and control (gray) and (b) grass (black) and control (gray), as well as statistically significant differences of experiments relative to control (shading, in Sv). Statistically insignificant differences are shown in white, see methods section. Insets in the bottom right of each panel show standard box plots of the annual mean AMOC index (in Sv) for (a) forest and control and (b) grass and control.

697 Seasonal mean meridional mass stream function (contours, every $2E10 \text{ kg s}^{-1}$) for (a,e) forest
698 (black) and control (gray) and (b,d) grass (black) and control (gray), as well as statistically
699 significant differences of both experiments relative to the control (shading, in $1E10 \text{ kg s}^{-1}$) for
700 (a,b) October to March and (c,d) April to September.

701 **Extended Figures**

Figure 4: **Land-area fraction Changes in the extratropical atmospheric circulation.** (a) Annual mean geopotential height at 500 hPa (Z500, shading, in %gpm) that is and meridional component of the eddy heat flux at 850 hPa ($\overline{av'T'}$, contours, -15, -10, 10, and 15 K ms^{-1}) covered with forest in the simulation control run, (b) afforested in the forest run annual mean frequencies of deep jets (forest minus control shading, in %) and E-vector divergence (b contours, -3, -1.5, 1.5, and 3 m^2s^{-2} (100 km deforested $^{-1}$), (c,e) differences in annual mean Z500 (shading, in gpm), eddy heat flux at 850 hPa (contours, -3, -1, 1, and 3 K ms^{-1}), and representative wind differences at 300 hPa (arrows, only the sixty globally largest differences at every tenth latitude and longitude index are shown, i.e. if larger than 0.6ms^{-1} for forest and 0.6ms^{-1} 1.4ms^{-1} int grass run) for (grass minus c) forest - control and (e) -In grass - control, and (ad,f) differences of annual mean deep jet frequencies (shading, grid in percent points [pp]) and E-vector divergence (contours, -0.15 and 0.15 m^2s^{-2} (100 km) $^{-1}$, only shown for regions where more E-vector divergence is larger than 50 % of the land area $0.5 \text{m}^2\text{s}^{-2}$ (100 km) $^{-1}$ in control is covered with grass) for (grey d) , shrubs forest - control and (green f) grass - control. For all contours, positive values are solid and erops negative values dashed. Statistically insignificant changes (blue see methods) are hatched for variables shown in control shading. For other variables, only statistically significant results are shown, except from the changes in E-vector divergence. These fields would otherwise be too patchy in grass and are not significant in forest according to the test used in this study.

Figure 5: Difference in annual Seasonal changes in the strength of the Hadley cell. Seasonal mean clear sky albedo at the Earth's meridional mass stream function (contours, every $2E10 \text{ kg s}^{-1}$) for (a,c) forest (black) and control (gray) and (b,d) grass (black) and control (gray), as well as differences of both experiments relative to the preindustrial control run (shading, in $1E10 \text{ kg s}^{-1}$) for (a,b) forest-October to March and (bc,d) grass-April to September. Statistically insignificant differences are hatched, see methods.

Difference in monthly mean surface radiative forcing (shading), clear sky surface albedo (purple contours, -0.3 and $+0.3$), and snow cover fraction (green contours, -25% and $+25\%$) for *forest-control*

Difference in monthly mean surface radiative forcing (shading), clear sky surface albedo (purple contour, -0.3 and $+0.3$), and snow cover fraction (-25% and $+25\%$) for *grass-control*

Standard box plots for different time periods in the three simulations showing the temporal evolution of global mean (a) 2 m temperature and (b) clear sky albedo. In (c), additionally the 50-year mean percentage of the Earth's surface covered with sea ice or snow is shown (lines).

Zonal mean vertical cross section of (a) annual mean temperature (shading, in K), zonal wind speed (contours, in ms^{-1} , with steps of 10 ms^{-1}) for control and (b,c) differences of annual mean temperature (shading, in K) and wind speed (contours, in ms^{-1} with steps of 0.5 ms^{-1} , transparent where not statistically robust), and the meridional temperature gradient (-0.25 and $0.25 \text{ K} (10E3\text{km})^{-1}$) for (b) forest minus control and (c) grass minus control.

Eddy kinetic energy at 300 hPa (contours, every 50 J/kg) for (a) forest (black) and control (grey); and (b) grass (black) and control (grey). Shading shows absolute differences for (a) forest-control and (b) grass-control.

Differences of the annual mean meridional heat transport over 40-year time intervals during the first 120 years of simulation (a,c,e) forest, and (b,d,f) grass with respect to the average of annual mean transport during the first 40 years in simulation control. The total heat transport (black) is separated into atmospheric (red) and ocean (blue) transport. Atmospheric heat transport is further split into moist (dotted) and dry (dashed) heat transport.

Climatological mean components of the atmospheric energy budget for control at a given latitude for (a) fluxes over land, (b) fluxes over the ocean, and (c) total fluxes. Negative values indicate net energy loss from the atmospheric column and positive values a net energy gain.

Absolute changes in the components of the atmospheric energy budget in forest at a given latitude averaged over 40-year time intervals during the first 120 years of the simulation forest with respect to the average over the first 40 years in control for (a,d,g) fluxes over land, (b,e,h) fluxes over the ocean, and (c,f,i) total fluxes. A reference for the climatological mean components of the annual

REVIEWERS' COMMENTS

Reviewer #3 (Remarks to the Author):

I appreciate the authors' great effort in addressing all my previous comments. Especially, the more in-depth analysis and detailed discussion of the thermodynamic changes and seasonality has improved the significance of the work. I have no further concerns and am pleased to recommend this revised manuscript for publication.

Global forestation and deforestation affect remote climate via adjusted atmosphere and ocean circulation

REPLIES TO THE REVIEWERS' COMMENTS

Raphael Portmann, Urs Beyerle, Edouard Davin, Erich M. Fischer, Steven De Hertog, Sebastian Schemm

REVIEWERS' COMMENTS

Reviewer #3 (Remarks to the Author):

I appreciate the authors' great effort in addressing all my previous comments. Especially, the more in-depth analysis and detailed discussion of the thermodynamic changes and seasonality has improved the significance of the work. I have no further concerns and am pleased to recommend this revised manuscript for publication.

We thank the reviewer for acknowledging our efforts and the positive evaluation of our manuscript.

[revised manuscript text omitted]
 ~~moisture~~ moist (dotted) and dry static energy (dashed). ~~The first 50 years of the simulations are excluded.~~

Figure 3: Slow-down and acceleration of the Atlantic meridional overturning circulation. Shown is the annual mean meridional mass stream function in the Atlantic, i.e. the AMOC (contours, in Sv, only values above 4 Sv shown with a 4 Sv interval) for (a) forest (black) and control (gray) and (b) grass (black) and control (gray), as well as differences of experiments relative to control (shading, in Sv). Statistically insignificant differences are shown in white, see methods section. Insets in the bottom right of each panel show standard box plots of the annual mean AMOC index (in Sv) for (a) forest and control and (b) grass and control.

Figure 4: Changes in the extratropical atmospheric circulation. (a) Annual mean geopotential height at 500 hPa (Z_{500} , shading, in gpm) and meridional component of the eddy heat flux at 850 hPa ($\overline{v'T'}$, contours, -15, -10, 10, and 15 K ms^{-1}) in simulation *control*, (b) annual mean frequencies of deep jets (shading, in %) and E-vector divergence (contours, -3, -1.5, 1.5, and 3 $\text{m}^2\text{s}^{-2}(100 \text{ km})^{-1}$), (c,e) differences in annual mean Z_{500} (shading, in gpm), eddy heat flux at 850 hPa (contours, -3, -1, 1, and 3 K ms^{-1}), and representative wind differences at 300 hPa (arrows, only the sixty globally largest differences at every tenth latitude and longitude index are shown, i.e. if larger than 0.6ms^{-1} for *forest* and 0.6ms^{-1} 1.4ms^{-1} int *grass*) for (c) *forest - control* and (e) *grass - control*, and (d,f) differences of annual mean deep jet frequencies (shading, in percent points [pp]) and E-vector divergence (contours, -0.15 and 0.15 $\text{m}^2\text{s}^{-2}(100 \text{ km})^{-1}$, only shown for regions where E-vector divergence is larger than $0.5 \text{m}^2\text{s}^{-2}(100 \text{ km})^{-1}$ in *control*) for (d) *forest - control* and (f) *grass - control*. For all contours, positive values are solid and negative values dashed. Statistically insignificant changes (see methods) are hatched for variables shown in shading. For other variables, only statistically significant results are shown, except from the changes in E-vector divergence. These fields would otherwise be too patchy in *grass* and are not significant in *forest* according to the test used in this study.

Figure 5: Seasonal changes in the strength of the Hadley cell. Seasonal mean meridional mass stream function (contours, every $2E10 \text{ kg s}^{-1}$) for (a,c) forest (black) and control (gray) and (b,d) grass (black) and control (gray), as well as differences of both experiments relative to the control (shading, in $1E10 \text{ kg s}^{-1}$) for (a,b) October to March and (c,d) April to September. Statistically insignificant differences are hatched, see methods.